# A mechanism in agrin signaling revealed by a prevalent Rapsyn mutation in congenital myasthenic syndrome

Guanglin Xing[1†], Hongyang Jing[1†], Lei Zhang[1], Yu Cao[2], Lei Li[1], Kai Zhao[1,2], Zhaoqi Dong[1], Wenbing Chen[1], Hongsheng Wang[1], Rangjuan Cao[1], Wen-Cheng Xiong[1,3], Lin Mei[1,3]*

[1]Department of Neurosciences, School of Medicine, Case Western Reserve University, Cleveland, United States; [2]Department of Neuroscience and Regenerative Medicine, Augusta University, Augusta, United States; [3]Louis Stokes Cleveland Veterans Affairs Medical Center, Cleveland, United States

**Abstract** Neuromuscular junction is a synapse between motoneurons and skeletal muscles, where acetylcholine receptors (AChRs) are concentrated to control muscle contraction. Studies of this synapse have contributed to our understanding of synapse assembly and pathological mechanisms of neuromuscular disorders. Nevertheless, underlying mechanisms of NMJ formation was not well understood. To this end, we took a novel approach – studying mutant genes implicated in congenital myasthenic syndrome (CMS). We showed that knock-in mice carrying N88K, a prevalent CMS mutation of Rapsyn (Rapsn), died soon after birth with profound NMJ deficits. Rapsn is an adapter protein that bridges AChRs to the cytoskeleton and possesses E3 ligase activity. In investigating how N88K impairs the NMJ, we uncovered a novel signaling pathway by which Agrin-LRP4-MuSK induces tyrosine phosphorylation of Rapsn, which is required for its self-association and E3 ligase activity. Our results also provide insight into pathological mechanisms of CMS.
DOI: https://doi.org/10.7554/eLife.49180.001

*For correspondence:
lin.mei@case.edu

†These authors contributed equally to this work

Competing interests: The authors declare that no competing interests exist.

## Introduction

The neuromuscular junction (NMJ) is a synapse between motoneurons and muscle fibers. At vertebrate NMJs, motor nerve terminals release acetylcholine (ACh), which activates ACh receptors (AChRs) concentrated on postsynaptic membranes to initiate muscle contraction. The concentration of AChR at postsynaptic membrane requires Agrin, a factor released from motoneurons (*McMahan, 1990*), which binds to LRP4 (*Kim et al., 2008*; *Zhang et al., 2008*) to activate MuSK (*DeChiara et al., 1996*; *Glass et al., 1996*; *Jennings et al., 1993*). Downstream effector of MuSK is believed to be Rapsn, a cytoplasmic protein that is required for AChR clustering and NMJ formation. Rapsn, discovered as a peripheral membrane protein associated with AChRs in the electric organ of Torpedo (*Cohen et al., 1972*; *Neubig and Cohen, 1979*; *Porter and Froehner, 1985*), colocalizes with AChRs at developing as well as adult NMJs (*Froehner et al., 1981*; *Noakes et al., 1993*; *Sealock et al., 1984*). It could induce AChR clusters in heterologous cells (*Froehner et al., 1990*; *Li et al., 2016*; *Phillips et al., 1991*); *Rapsn* null mutant mice die soon after birth without AChR clusters (*Gautam et al., 1995*), indicating a critical role in NMJ formation. Being a classic adapter protein, Rapsn is thought to bridge the AChR to the cytoskeleton (*Apel et al., 1995*; *Bartoli et al., 2001*; *Chen et al., 2016*; *Lee et al., 2009*; *Maimone and Merlie, 1993*; *Miyazawa et al., 1999*; *Ramarao et al., 2001*; *Ramarao and Cohen, 1998*). We showed recently that Rapsn possesses E3 ligase activity (*Li et al., 2016*). Mutation of a cysteine residue necessary for the E3 ligase activity

impairs its ability to cluster AChRs in vitro and to form NMJs in knock-in mice, suggesting that Rapsn may also be a signaling molecule. Nevertheless, how signal is transduced from upstream molecules such as Agrin-LRP4-MuSK to Rapsn, remains unclear, a glaring gap in our understanding of NMJ formation.

Congenital myasthenic syndromes (CMSs) are a heterogeneous group of NMJ diseases caused by mutations of genes of NMJ structure and function proteins (*Engel et al., 2003*; *Engel et al., 2015*; *Engel and Sine, 2005*; *Hantaï et al., 2004*). Based on the primary deficits, CMSs could be classified into presynaptic, postsynaptic, and synaptic cleft-associated groups (*Engel et al., 2003*; *Engel et al., 2015*; *Engel and Sine, 2005*). A majority of CMS cases, ~75%, involve mutations of genes encoding proteins for postsynaptic development or function; and among them, mutations in AChR subunits and Rapsn are prevalent (*Engel et al., 2003*; *Engel et al., 2015*; *Engel and Sine, 2005*). Although more than thirty genes have been identified in CMS patients, mouse models to mimic such mutations are rare. Hence underlying pathological mechanisms remain not well understood. Presently, about 40 mutations in the *Rapsn* gene have been identified in patients with CMSs, accounting for 15% of total CMS cases (*Dunne and Maselli, 2003*; *Maselli et al., 2007*; *Maselli et al., 2003*; *Milone et al., 2009*; *Müller et al., 2004*; *Ohno and Engel, 2004*; *Ohno et al., 2002*; *Ohno et al., 2003*; *Yasaki et al., 2004*). Of them, the N88K (asparagine 88 to lysine) mutation is the most common (*Milone et al., 2009*; *Müller et al., 2003*). About 90% of Rapsn-related CMS patients carry N88K mutation (*Burke et al., 2003*; *Milone et al., 2009*; *Müller et al., 2003*). Patients are either homozygous for N88K or heteroallelic for N88K with another *Rapsn* mutation. Pathological examination of patient specimens revealed attenuated or fragmented AChR clusters and reduced Rapsn protein, and altered junctional folds (*Maselli et al., 2003*; *Milone et al., 2009*; *Ohno et al., 2002*). In severe cases, N88K heteroallelic with c.966 + 1GT, L14P, or a frameshift mutation may cause postnatal death (*Maselli et al., 2003*; *Milone et al., 2009*; *Richard et al., 2003*). N88K mutant Rapsn induced fewer AChR clusters, compared with wild type Rapsn in heterologous cells (*Ohno et al., 2002*) and was unable to fully rescue Agrin-induced AChR clusters in *Rapsn* mutant cultured myotubes (*Cossins et al., 2006*).

To understand molecular mechanisms of NMJ formation, we studied CMS mutations of the *Rapsn* gene with an idea that such mutations with clinic implications would reveal unexpected insight. In particular, we generated N88K knock-in mice to investigate the in vivo impact of N88K mutation on the NMJ. N88K mutant mice displayed profound deficits not only in AChR clusters, but also in nerve terminals at both light and electron microscopic levels. Developmentally, the NMJ abnormality was observed as early as embryonic day 14. We performed a combination of cell and molecular biological experiments including studying N88K mutant muscles. We demonstrated that N88K mutation inhibited the E3 ligase activity of Rapsn by reducing its phosphorylation and self-association. Our results not only unravel pathophysiological mechanisms of N88K mutation, but also provide novel insight into how signals are transduced from Agrin-LRP4-MuSK to Rapsn.

## Results

### Aberrant NMJ formation in N88K mt mice

To study pathological mechanisms of the CMS-associated mutation N88K, we generated N88K knock-in mutant (mt) mice by CRISPR-Cas9 technique (*Figure 1—figure supplement 1A and B*). The mRNA of N88K *Rapsn* was comparable to that of wild type (WT) *Rapsn* (*Figure 1—figure supplement 1C*). The protein levels of Rapsn were similar between muscle lysates of the two genotypes (*Figure 1A*); and the mutation seemed to have little effect on the levels of ubiquitinated Rapsn (*Figure 1—figure supplement 1D and E*). However, N88K homozygous mt mice died within 24 hr of birth with cyanosis, suggesting respiratory failure. To examine the effect of N88K mutation on the NMJ, diaphragms were isolated from neonatal P0 mice and stained for AChR and nerve terminals. In WT diaphragms, AChR clusters were abundant and localized in the middle of muscle fibers, whereas in *Rapsn* null mt (-/-) mice, there was almost no AChR cluster (*Figure 1B–D*). N88K mt mice were able to form AChR clusters, but their number and size were dramatically reduced (*Figure 1C–F* and *Figure 1—figure supplement 1F*). Remaining clusters were elongated, in contrast to oval plaque morphology in WT mice (*Figure 1D and G* and *Figure 1—figure supplement 1G*) and distributed in a wider region of muscle fibers (*Figure 1C and H*). AChR clusters in N88K mt mice were reduced in

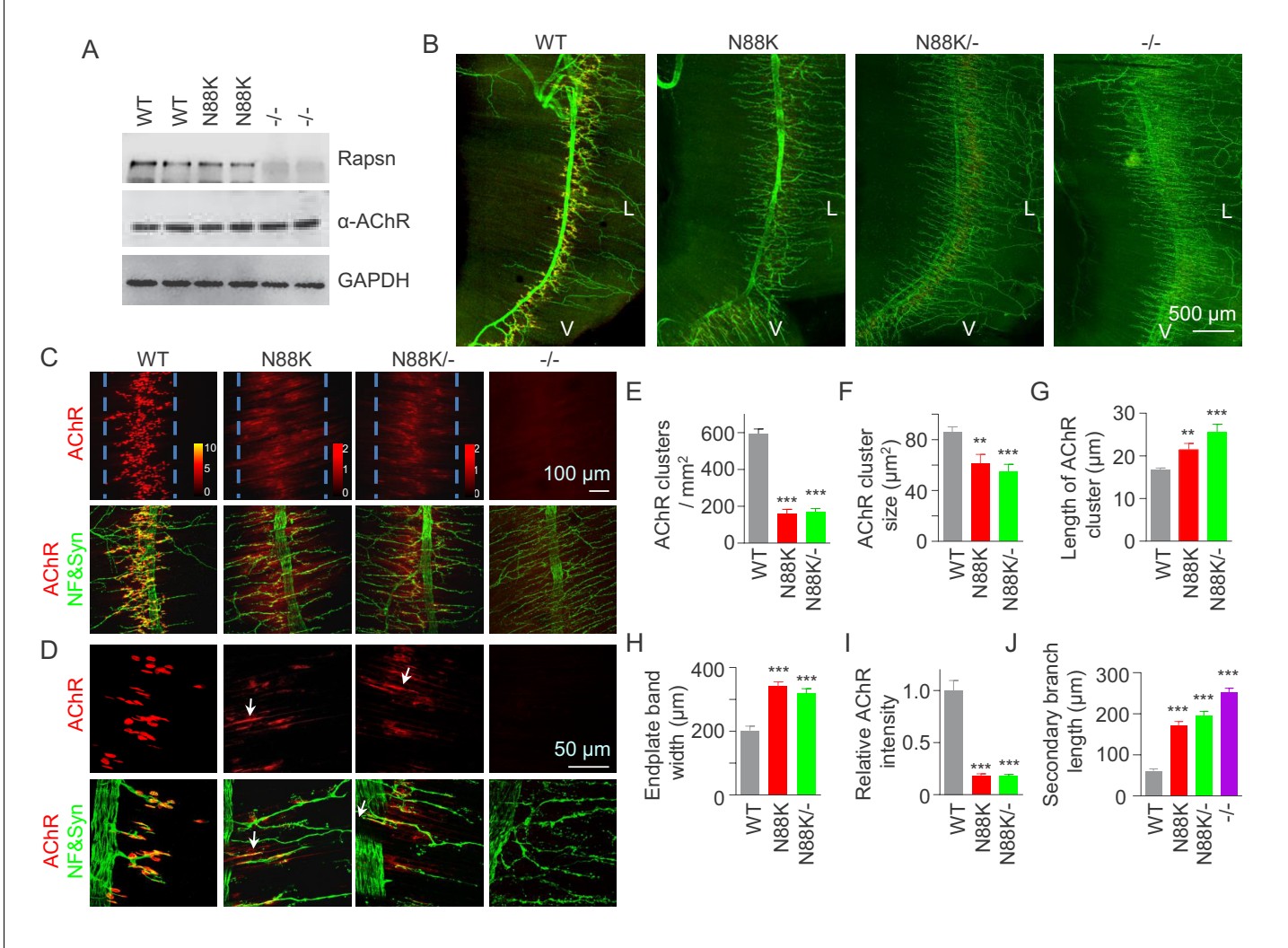

**Figure 1.** Few AChR clusters and extensive nerve terminal arborization in N88K mt mice. (**A**) Comparable Rapsn protein level between WT and N88K mt mice. Tissue lysates from WT, N88K mt, -/- (*Rapsn* null mt) diaphragms were subjected to western blotting with anti-Rapsn, and anti-α-AChR antibodies, using GAPDH as loading control. Note that α-AChR protein level was not altered in N88K and -/-, compared with WT controls. (**B**) Reduced AChR clusters and extensive axonal arborization in N88K mt mice. P0 diaphragms from WT, N88K, N88K/-, -/- mice were stained whole-mount with Flour 594-α-BTX (red) to label AChR clusters and with anti-NF/Syn antibodies (green) to label motor nerve terminals. V, ventral. L, left. (**C, D**) Higher magnification views of AChR clusters and motor axon branches. The boundary of AChR clusters was indicated by dashed blue lines. Heatmap, AChR intensity; arrows, elongated AChR clusters along axons. (**E–J**) Quantitative analysis of AChR cluster number (**E**), AChR cluster size (**F**), AChR cluster length (**G**), endplate band width (**H**), AChR fluorescence intensity (**I**), secondary branch length (**J**). Data were shown as mean ± SEM; **, p<0.01; ***, p<0.001, One-way ANOVA, n = 6 mice. Also see *Figure 1—figure supplement 1*.
DOI: https://doi.org/10.7554/eLife.49180.002

The following source data and figure supplements are available for figure 1:

**Source data 1.** Sample size (n), mean, SEM, p value, statistical methods and results are presented in *Figure 1E–J*.
DOI: https://doi.org/10.7554/eLife.49180.005
**Figure supplement 1.** Reduced AChR cluster size, increased AChR cluster length and extensive nerve terminal arborization in N88K mt mice.
DOI: https://doi.org/10.7554/eLife.49180.003
**Figure supplement 1—source data 1.** Raw data, sample size (n), mean, SEM, p value, statistical methods and results are presented in *Figure 1—figure supplement 1C, E and H*.
DOI: https://doi.org/10.7554/eLife.49180.004

staining intensity (*Figure 1I*), indicating reduced AChR concentration. In addition, N88K mt mice displayed abnormal nerve terminal arborization, with increased numbers of secondary, tertiary, quaternary branches, as observed in null mt mice (*Figure 1B–D and J* and *Figure 1—figure supplement 1H*). To eliminate potential off-target effects of CRISPR-Cas9, we generated N88K/- mice that possess null mutation on one chromosome and N88K mutation on the other. They exhibited similar NMJ deficits to N88K mt mice (*Figure 1B–J* and *Figure 1—figure supplement 1F–H*), suggesting that NMJ deficits in N88K mt mice are likely due to N88K mutation, not an off-target effect.

At electron microscopic (EM) level, in WT mice, axon terminals were filled with abundant synaptic vesicles, some of which were clustered at active zones (*Figure 2A and C*). Synaptic clefts were filled with synaptic basal lamina; and, on the postsynaptic side, junctional folds were observable (*Figure 2A and C*). However, junctional folds in N88K mt mice were fewer and the remainders were shorter than those of controls (*Figure 2B,D and E*). N88K mt mice also showed reduced number of synaptic vesicles in axon terminals (*Figure 2F*), compared with WT controls. The vesicle diameters and synaptic cleft width were similar between two genotypes (*Figure 2G and H*). Taken together, studies of P0 mice revealed both pre- and post- deficits at the NMJ, providing a mechanism of neonatal lethality of N88K mutation.

Muscles form primitive, aneural AChR clusters in advance of the arrival of motor nerve terminals (*Lin et al., 2001*; *Yang et al., 2001b*; *Luo, 2010*). Innervation induces the formation of new AChR clusters, perhaps by enlarging some of the aneural clusters. The formation of aneural clusters requires LRP4, MuSK, and Rapsn, but not Agrin (*Lin et al., 2001*; *Remédio et al., 2016*; *Yang et al., 2001b*). Having observed NMJ deficits in neonatal mice, we wondered whether the N88K mutation alters aneural AChR clusters and dissected diaphragms from embryos at E14, when axon terminals start branching out to innervate muscle fibers (*Li et al., 2018*; *Lin et al., 2001*; *Wu et al., 2010*; *Yang et al., 2001b*). As shown in *Figure 2I and K*, aneural AChR clusters of WT mice were numerous and elongated in shape (*Figure 2M and N*). Most of them were not innervated although nerve terminals were present. Remarkably, aneural AChR clusters were rarely detectable in N88K mt diaphragms (*Figure 2J,L,M and N*). These results suggest that N88 is necessary for the formation of aneural clusters and that the AChR clusters at P0 N88K mt diaphragms are likely induced by nerve terminals.

To examine whether synaptic transmission is altered in N88K mt mice, we first recorded resting membrane potentials and found that they were comparable between WT and N88K mt mice (*Figure 3A*). Next, we measured miniature endplate potentials (mEPP), postsynaptic potentials elicited by spontaneous vesicle release. In WT controls, mEPPs were easily detectable in ~88% of muscle fibers; however, this number was reduced to ~13% in N88K homozygous mt mice (*Figure 3B*). In addition, mEPP amplitudes and frequencies were reduced in N88K mt mice, compared with WT controls (*Figure 3C–E*). While N88K heterozygous (i.e., N88K/+) mice showed no detectable mEPP deficits (*Figure 3*), those in N88K/- and N88K homozygous mice were comparable (*Figure 3*). These results are in agreement with morphological deficits and indicate neuromuscular transmission is impaired by N88K mutation.

## Impaired ability of N88K Rapsn in AChR clustering

Rapsn is able to induce AChR clusters in heterologous cells (*Froehner et al., 1981*; *Li et al., 2016*; *Phillips et al., 1991*). To investigate mechanisms of how N88K mutation reduced AChR concentration at the NMJ, HEK293T cells were transfected with EGFP-tagged WT or N88K Rapsn together with four different AChR subunits (α, β, γ, δ), fixed and stained for AChR clusters. As shown in *Figure 4A*, AChRs were diffused when coexpressed with EGFP empty vector, but became clustered in cells co-expressing WT Rapsn, and these AChR clusters colocalized with Rapsn clusters, in agreement with previous reports (*Froehner et al., 1990*; *Li et al., 2016*; *Phillips et al., 1991*). N88K was able to form aggregates in HEK293 cells (*Figure 4A*), in agreement with a previous report (*Ohno et al., 2002*). However, fewer AChR clusters were observed in HEK293T cells expressing N88K, compared with those expressing WT Rapsn (*Figure 4A and B*). This result could suggest that the N88K mutation prevents Rapsn from clustering AChRs in HEK293T cells or maintaining AChR surface expression. To determine whether N88K mutation alters surface AChR levels, HEK293T cells were incubated with Sulfo-NHS-SS-biotin at 4℃. Biotin-labeled surface proteins were precipitated and probed for different AChR subunits and transferrin (Trf) as control. As shown in *Figure 4C*, surface AChR, total AChR and Rapsn levels were comparable between cells transfected with WT and

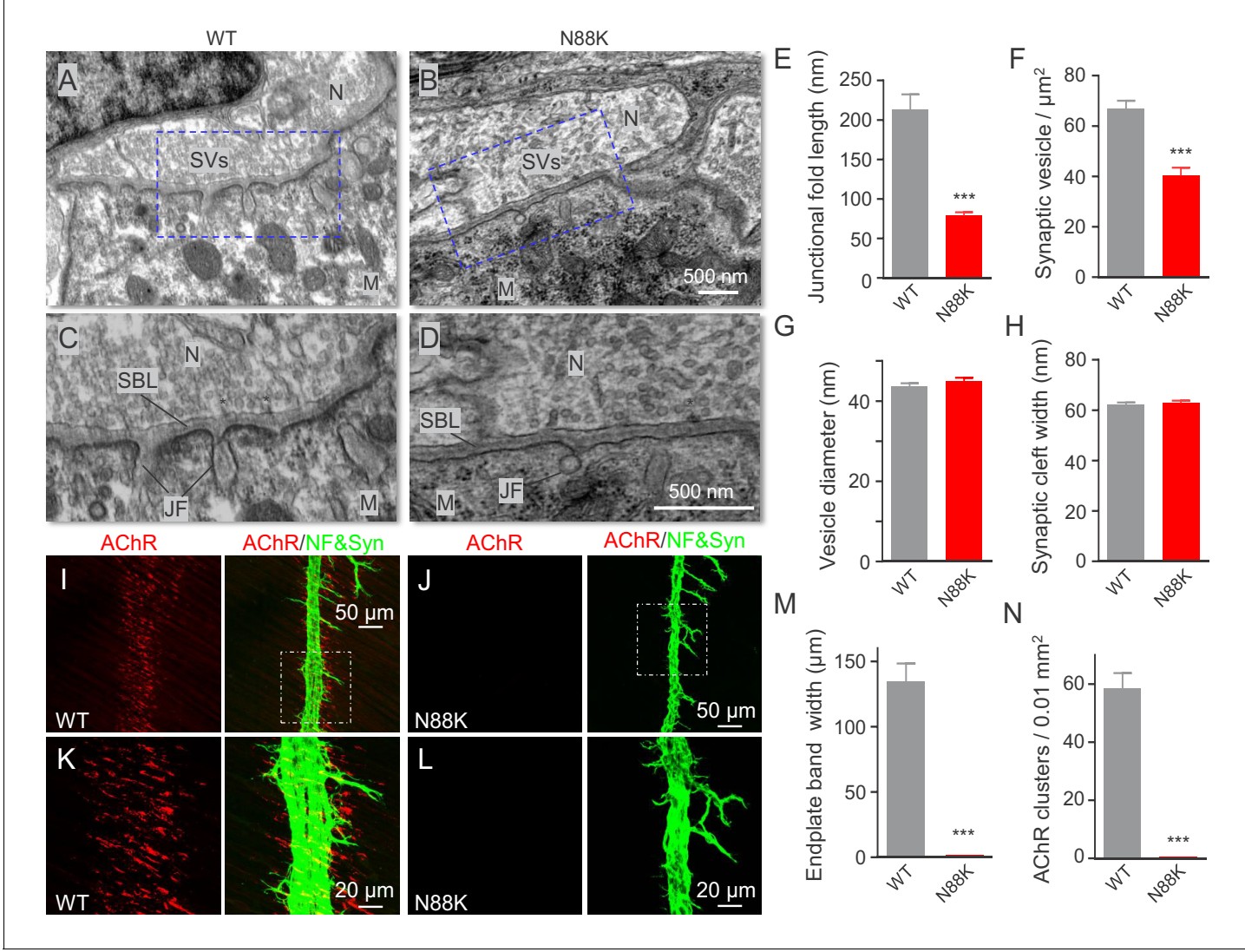

**Figure 2.** Small junctional folds, reduced vesicle density and diminished aneural AChR clusters in N88K mt mice. (**A, B**) Low magnification NMJ EM images of WT and N88K mt, including synaptic vesicles (SVs), synaptic cleft, synaptic basal lamina (SBL), and postsynaptic junctional folds (JF). M, muscle; N, nerve; (**C, D**) High magnification images of SVs, SBL, and JF. Note that junctional folds were smaller in N88K mt, and vesicle density was reduced in N88K mt, compared with WT controls. Asterisks indicate active zone. (**E–H**) Quantitative data of junctional fold length (**E**), synaptic vesicle density (**F**), vesicle diameter (**G**), synaptic cleft width (**H**) of WT and N88K mt. Data were shown as mean ± SEM; ***, p<0.001, unpaired t-test, n = 3 mice. (**I–L**) Few aneural AChR clusters in N88K mt mice at E14. E14 diaphragms from WT, N88K mt mice were stained whole-mount with Flour 594-α-BTX (red) to label AChR clusters and with anti-NF/Syn (green) antibodies to label motor nerve terminals. (**I, J**) Low magnification view. (**K, L**) High magnification view. (**M, N**) Quantitative data of endplate band width and AChR cluster number. Data were shown as mean ± SEM, ***, p<0.001, unpaired t-test, n = 3 mice.

DOI: https://doi.org/10.7554/eLife.49180.006

The following source data is available for figure 2:

**Source data 1.** Sample size (n), mean, SEM, p value, statistical methods and results are presented in *Figure 2E, F, G, H, M and N*.
DOI: https://doi.org/10.7554/eLife.49180.007

N88K Rapsn, suggesting that N88K mutation did not alter AChR levels on cell surface. Moreover, N88K mutation did not change Rapsn's stability in HEK293T cells (*Figure 4—figure supplement 1A and B*). These results suggest that the ability of N88K mt Rapsn to induce AChR clusters was reduced.

Next, to test the ability of N88K mt Rapsn in AChR clustering in muscle cells, we generated N88K mt C2C12 cells using the same CRISPR-Cas9 strategy to generate N88K mt mice. The mutation was

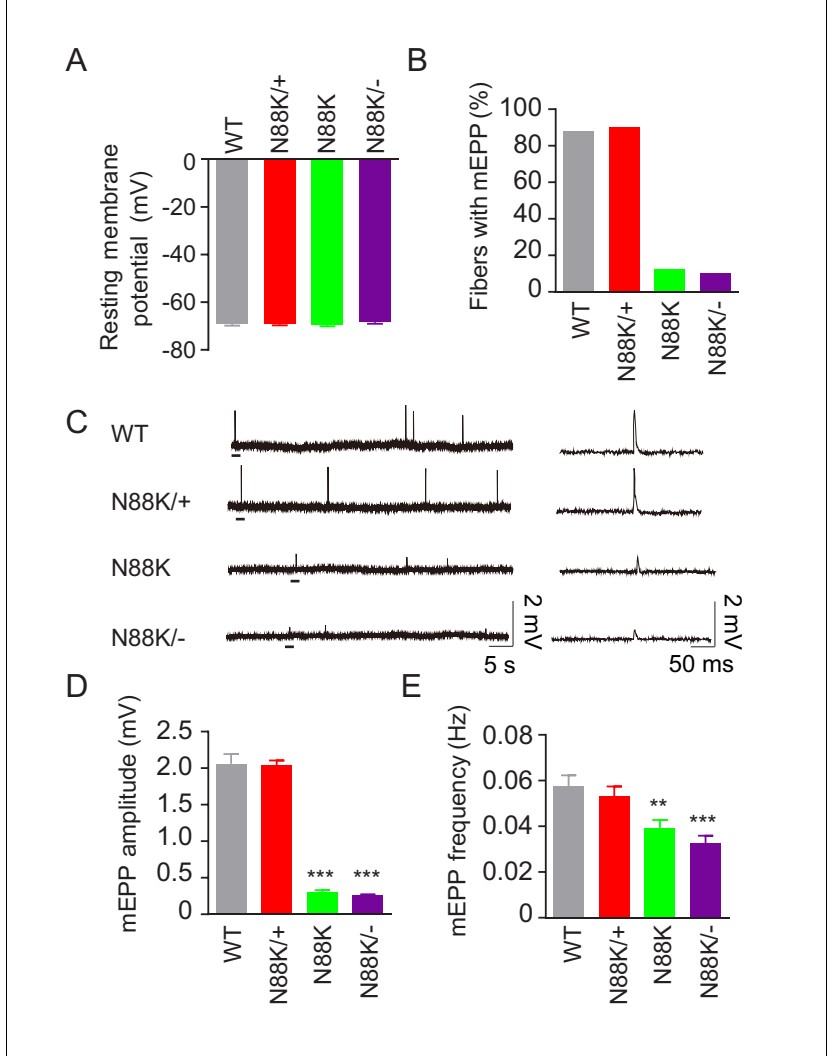

**Figure 3.** Reduced synaptic transmission in N88K mt mice. (**A**) Comparable resting membrane potentials among WT, N88K/+, N88K mt, and N88K/- mt. P0 hemi-diaphragms at ventral, left were recorded. (**B**) Percentage of muscle fibers with mEPPs in 3 min of recording (n = 120, 6 mice). (**C**) Representative mEPP traces. (**D, E**) Reduced mEPP amplitude (**D**) and reduced mEPP frequency in N88K mt and N88K/- mt (**E**). Data were shown as mean ± SEM, **, p<0.01, ***, p<0.001, One-way ANOVA, n = 6 mice.

DOI: https://doi.org/10.7554/eLife.49180.008

The following source data is available for figure 3:

**Source data 1.** Sample size (n), mean, SEM, p value, statistical methods and results are presented in *Figures 3A, B, D and E*.

DOI: https://doi.org/10.7554/eLife.49180.009

---

confirmed by genomic DNA sequencing (*Figure 4—figure supplement 2A*). The levels of N88K mRNA (*Figure 4—figure supplement 2B*) and protein (*Figure 4D*) in mt C2C12 myotubes were comparable to those of WT Rapsn in control myotubes. The mutation had no apparent effect on the stability of Rapsn protein (*Figure 4—figure supplement 2C and D*) and surface AChR levels (*Figure 4D*). However, AChR clusters were reduced in Agrin-treated N88K mt C2C12 myotubes, in contrast to robust AChR clusters in Agrin-treated WT C2C12 myotubes (*Figure 4E and F*). These results demonstrated that N88K mutation impairs the ability of Rapsn to induce AChR clusters.

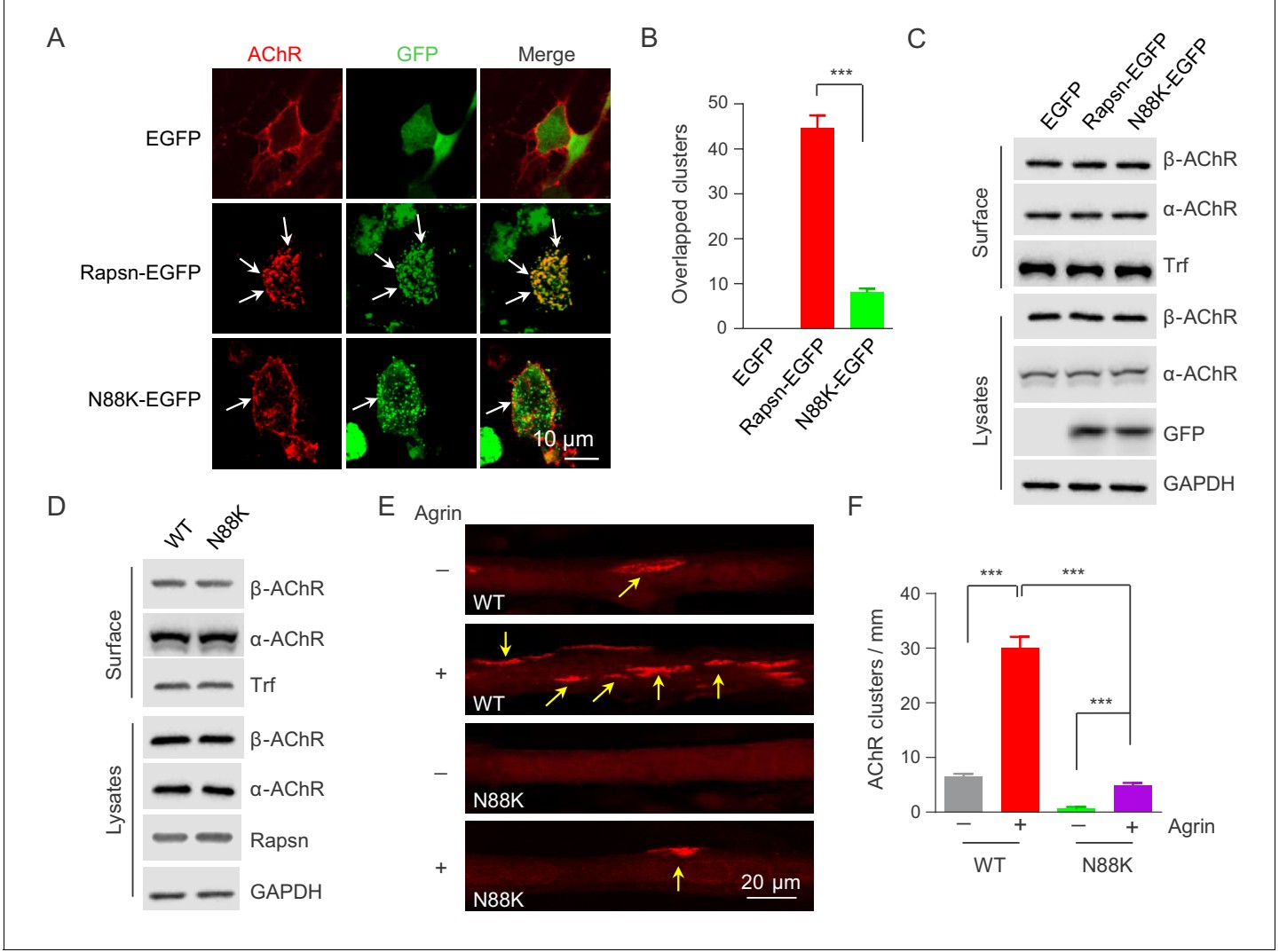

**Figure 4.** Impaired ability of N88K mt Rapsn in AChR clustering in HEK293T cells and in cultured muscle cells. (A) Impaired ability of N88K mt Rapsn to induce AChR clustering in HEK293T cells. HEK293T cells were transfected with AChR subunits (α, β, γ, δ), along with EGFP empty vector, Rapsn-EGFP, or N88K-EGFP. After 36 hr, live, unfixed cells were incubated with Flour 594-α-BTX (red) to label surface AChRs. Arrows, AChR clusters colocalized with Rapsn clusters. (B) Quantitative data of (A) (mean ± SEM). ***, p<0.001, unpaired t-test, n = 20 cells. (C) Comparable amount of AChRs and Rapsn in (A). Total levels of α-AChR, β-AChR, Rapsn, and surface α-AChR, β-AChR in the parallel experiment of (A) were examined by western blotting, using GAPDH and Transferrin (Trf) as lysate and surface protein loading controls, respectively. (D) Comparable total Rapsn and AChR, and surface AChR expression in N88K mt cultured myotubes. Total α-AChR, β-AChR, Rapsn and surface α-AChR, β-AChR from WT or N88K mt C2C12 myotubes were examined by western blotting, using GAPDH and Trf as lysate and surface protein loading controls, respectively. (E) Fewer Agrin-induced AChR clusters in N88K mt C2C12 myotubes, compared with WT controls. Myotubes were treated with or without 50 ng/ml Agrin for 12 hr. Arrows, AChR clusters. (F) Quantitative data of (E) (mean ± SEM), ***, p<0.001, Two-way ANOVA, n = 20 cells. Also see *Figure 4—figure supplement 1* and *Figure 4—figure supplement 2*.

DOI: https://doi.org/10.7554/eLife.49180.010

The following source data and figure supplements are available for figure 4:

**Source data 1.** Sample size (n), mean, SEM, p value, statistical methods and results are presented in *Figure 4B and F*.
DOI: https://doi.org/10.7554/eLife.49180.015

**Figure supplement 1.** Comparable protein stability between WT and N88K mt Rapsn in HEK293T cells.
DOI: https://doi.org/10.7554/eLife.49180.011

**Figure supplement 1—source data 1.** Raw data, sample size (n), mean, SEM, p value, statistical methods and results are presented in *Figure 4—figure supplement 1B*.
DOI: https://doi.org/10.7554/eLife.49180.012

**Figure supplement 2.** Comparable *Rapsn* mRNA level and Rapsn protein stability between WT and N88K mt myotubes.

*Figure 4 continued on next page*

*Figure 4 continued*

DOI: https://doi.org/10.7554/eLife.49180.013

**Figure supplement 2—source data 1.** Raw data, sample size (n), mean, SEM, p value, statistical methods and results are presented in *Figure 4—figure supplement 2B and D*.

DOI: https://doi.org/10.7554/eLife.49180.014

## Reduced E3 ligase activity of N88K mt Rapsn

As a scaffold protein, Rapsn could induce AChR clusters by bridging AChR subunits to the actin cytoskeleton (*Burden et al., 1983*; *LaRochelle and Froehner, 1986*; *Zhang et al., 2007*; *Walker et al., 1984*). Therefore, we first determined whether the N88K mutation alters Rapsn binding to surface AChRs and cytoplasmic actin. Surface AChRs were purified with biotin-α-BTX/Streptavidin beads and probed for Rapsn and actin (*Figure 5A*). As shown in *Figure 5B*, the amount of Rapsn and actin associated with surface AChRs was similar between WT and N88K mt myotubes. When co-expressed in HEK293 cells, AChR subunits precipitated by WT or N88K mt Rapsn were similar (*Figure 5—figure supplement 1A and B*), indicating the mutation has little effect on the binding to AChRs. Actin-anchored Rapsn-AChR complexes are resistant to low concentration of detergents (*Moransard et al., 2003*). We found that the amount of Rapsn that could be solubilized by Triton X-100 was similar between N88K mt and WT myotubes (*Figure 5—figure supplement 1C and D*). These results suggest that the N88K mutation may not impair Rapsn association with actin cytoskeleton components.

Our recent study revealed that Rapsn possesses E3 ligase activity, which is required for AChR clustering and NMJ formation (*Li et al., 2016*). In a working model, Rapsn increases AChR clusters by enhancing AChR neddylation, a posttranslational modification with Nedd8 (an Ub-like molecule) (*Li et al., 2016*; *Li et al., 2018*). To examine whether N88K mutation impairs Rapsn-mediated neddylation of AChRs, HA-tagged WT or N88K mt Rapsn was co-transfected with Flag-tagged δ-AChR and Myc-tagged Nedd8 into HEK293T cells. δ-AChR was purified by anti-Flag antibody and probed with anti-Nedd8 antibody. Nedd8-conjugated δ-AChR was apparent in HEK293T cells co-expressing WT Rapsn (*Figure 5C and D*); however, Nedd8 signaling was dramatically reduced in cells co-expressing N88K Rapsn (*Figure 5C and D*), to a level of cells expressing C366A, an E3 ligase-dead *Rapsn* mt (*Li et al., 2016*). These results suggest that N88K mutation may impair the E3 ligase activity of Rapsn. To further test this hypothesis, we compared neddylated δ-AChR level between WT and N88K mt C2C12 myotubes and found that neddylated δ-AChR in N88K mt myotubes was remarkably reduced, compared with WT controls (*Figure 5E and F*). Finally, we examined neddylated δ-AChR level in skeletal muscles of WT and N88K mt mice. As shown in *Figure 5G and H*, neddylated δ-AChR level was reduced in muscles of N88K mt mice, compared with WT controls. Together these results suggest that impaired E3 ligase activity may be a mechanism of N88K mutation.

## Reduced Rapsn tyrosine phosphorylation by N88K mutation

Agrin binds to LRP4 to stimulate MuSK to induce AChR clustering. How signal is transduced from MuSK to Rapsn was not well understood. Rapsn was tyrosine phosphorylated in electric organs of Torpedo California where AChRs are enriched (*Mohamed and Swope, 1999*) and in heterologous cells when expressed alone or together with MuSK (*Lee et al., 2008*). We characterized Rapsn tyrosine phosphorylation in muscle cells in response to Agrin treatment. Rapsn and MuSK were precipitated with respective antibodies from C2C12 myotubes treated with Agrin for different times and probed with anti-p-Tyr antibody. As shown in *Figure 6A*, MuSK activation occurred within 10 min, which was followed by Rapsn tyrosine phosphorylation. On the other hand, the increased in δ-AChR neddylation, an indicator of Rapsn E3 ligase activity (*Li et al., 2016*), was not peaked until 90 min after Agrin stimulation (*Figure 6B*). These data, quantified in *Figure 7J*, suggest that these events occur sequentially.

To determine whether N88K mutation alters Rapsn tyrosine phosphorylation in muscle cells, we studied N88K mt C2C12 myotubes together with WT and *Rapsn* null (-/-) C2C12 myotubes. Compared with WT C2C12 myotubes, Agrin-stimulated Rapsn tyrosine phosphorylation in N88K mt myotubes was dramatically reduced (*Figure 6C and D*). These results demonstrate that Rapsn becomes

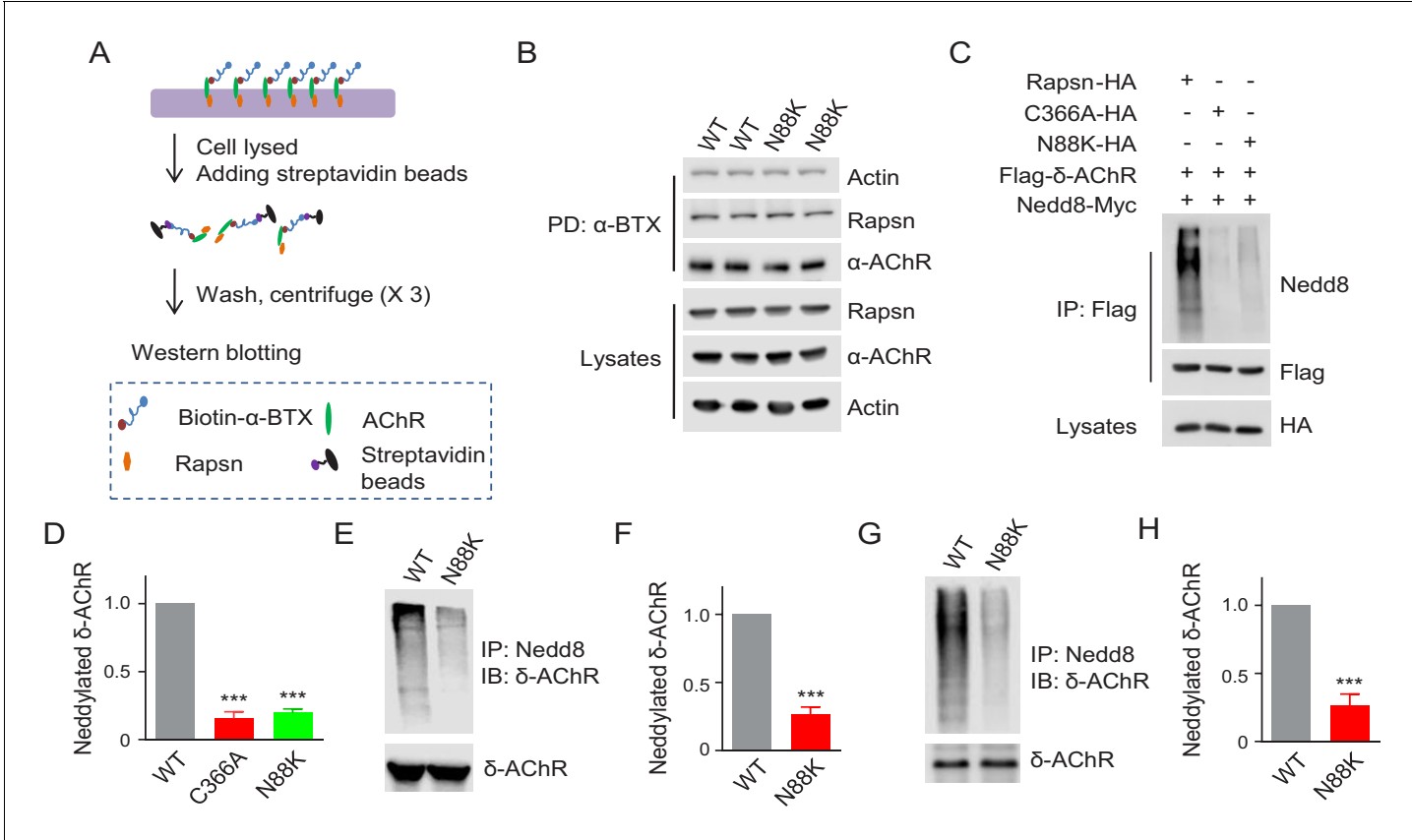

**Figure 5.** Reduced E3 ligase activity in N88K mt Rapsn. (**A**) Schematic diagram of extraction of surface AChRs from C2C12 myotubes. Live C2C12 myotubes were incubated with biotin-α-BTX at 4°C for 1 hr to capture AChR complex, and then were lysed. Resulting biotin-α-BTX-AChR complex in lysates were precipitated by streptavidin-coupled agarose beads. Cell lysates, precipitated AChRs and AChR-associated proteins were examined by western blotting. (**B**) Comparable amounts of Actin and Rapsn were co-precipitated by surface AChR between WT and N88K mt C2C12 myotubes. Surface AChRs from WT or N88K mt myotubes were isolated, and associated Rapsn and actin were examined by western blotting. (**C**) Reduced Rapsn E3 ligase activity by N88K mutation in HEK293T cells. HEK293T cells were transfected with HA tagged WT, C366A, or N88K mt Rapsn, along with Flag-δ-AChR and Nedd8-Myc. After 48 hr, cells were lysed and precipitated with anti-Flag beads to pull down δ-AChR. The precipitated δ-AChR was blotted with anti-Nedd8 antibody to examine its neddylation. (**D**) Quantitative data of neddylated δ-AChR in (**C**) (mean ± SEM), ***, p<0.001, One-way ANOVA, n = 3. (**E–H**) Reduced Rapsn E3 ligase activity in N88K mt cultured myotubes and in mt mice. (**E, F**) WT and N88K mt cultured myotubes were treated with Agrin for 2 hr. Myotubes were lysed and precipitated with anti-Nedd8 antibody. The resulting lysates and precipitated proteins were blotted with indicated antibodies to reveal neddylated δ-AChR, readout for E3 ligase activity of Rapsn (**E**). (**F**) Quantitative data in (**E**) (mean ± SEM), ***, p<0.001, unpaired t-test, n = 3. (**G**) Neddylated δ-AChR was examined in WT or N88K mt mice. (**H**) Quantitative data in (**G**) (mean ± SEM), ***, p<0.001, unpaired t-test, n = 3. Also see *Figure 5—figure supplement 1*.

DOI: https://doi.org/10.7554/eLife.49180.016

The following source data and figure supplements are available for figure 5:

**Source data 1.** Raw data, sample size (n), mean, SEM, p value, statistical methods and results are presented in *Figure 5D, F and H*.
DOI: https://doi.org/10.7554/eLife.49180.019

**Figure supplement 1.** Comparable binding ability of N88K mt Rapsn with subunits of AChR and Actin.
DOI: https://doi.org/10.7554/eLife.49180.017

**Figure supplement 1—source data 1.** Raw data, sample size (n), mean, SEM, p value, statistical methods and results are presented in *Figure 5—figure supplement 1D*.
DOI: https://doi.org/10.7554/eLife.49180.018

tyrosine phosphorylated in response to Agrin stimulation and this event is inhibited by N88K mutation. To determine whether N88K mutation alters Rapsn tyrosine phosphorylation in vivo, Rapsn protein was precipitated with anti-Rapsn antibody from muscle homogenates of mice at E14 and P0 and probed with anti-p-Tyr antibody (*Figure 6E–G*). Rapsn tyrosine phosphorylation from N88K mt mice

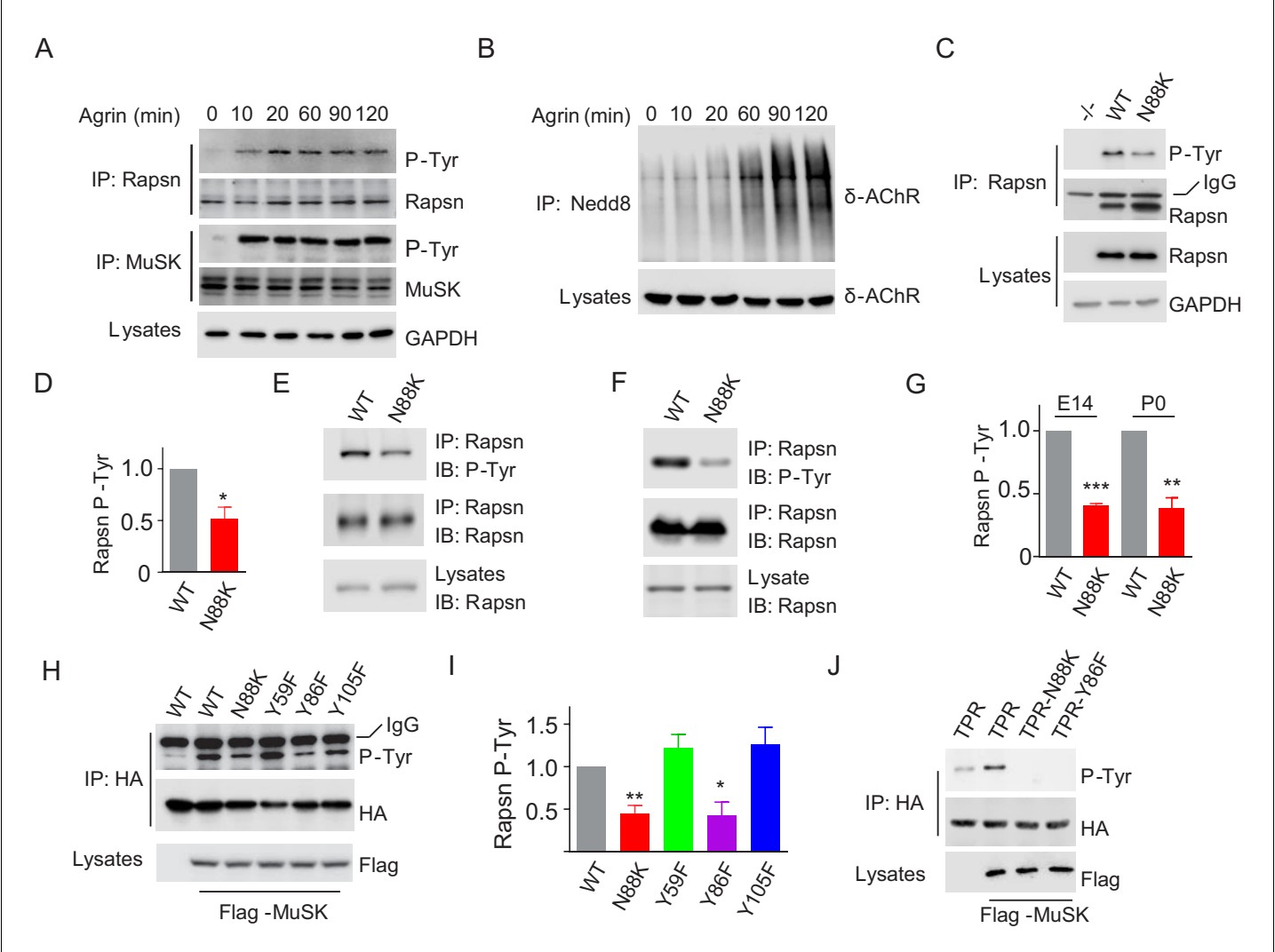

**Figure 6.** Impaired Y86 phosphorylation by N88K mutation. (**A**) Agrin treatment induced tyrosine phosphorylation of Rapsn in cultured myotubes. WT cultured myotubes were treated with Agrin for indicated times. Cells were lysed and were incubated with anti-Rapsn antibody to precipitate Rapsn protein, and probed with anti-p-Tyr antibody to examine tyrosine phosphorylation of Rapsn. MuSK tyrosine phosphorylation was examined as positive controls. (**B**) Agrin treatment induced E3 ligase activity of Rapsn in culture myotubes. WT cultured myotubes were treated with Agrin for indicated times. Neddylated δ-AChR was examined to reveal E3 ligase activity of Rapsn. (**C**) Reduced tyrosine phosphorylation of Rapsn by N88K mutation in cultured myotubes. WT, N88K mt, *Rapsn* null mt (-/-) cultured myotubes were treated with Agrin for 2 hr, and then tyrosine phosphorylation of Rapsn was examined. (**D**) Quantitative data in (**C**) (mean ± SEM), *, p<0.05, unpaired t-test, n = 3. (**E–G**) Reduced tyrosine phosphorylation of Rapsn by N88K mutation in mt mice. Rapsn protein was precipitated with anti-Rapsn antibody from muscle homogenates of WT or N88K mice at E14 (**E**) and P0 (**F**) and probed with anti-p-Tyr antibody to examine tyrosine phosphorylation of Rapsn. (**G**) Quantitative data in (**E**) and (**F**) (mean ± SEM), **, p<0.01, ***, p<0.001, unpaired t-test, n = 3. (**H**) Reduced MuSK-induced Rapsn tyrosine phosphorylation by N88K and Y86F mutations in HEK293T cells. HEK293T cells were transfected HA tagged WT or indicated mt Rapsn, along with Flag tagged MuSK, or not. Precipitated Rapsn protein with anti-HA beads was probed with anti-p-Tyr antibody to reveal tyrosine phosphorylation of Rapsn. (**I**) Quantitative data of (**H**) (mean ± SEM), *, p<0.05, **, p<0.001, unpaired t-test, n = 3. (**J**) Abolished MuSK-induced TPR1-7 (TPR) tyrosine phosphorylation by N88K and Y86F mutations in HEK293T cells. Same as (**H**), MuSK-induced WT, Y86F, or N88K mt TPR was detected in HEK293T cells.

DOI: https://doi.org/10.7554/eLife.49180.020

The following source data is available for figure 6:

**Source data 1.** Raw data, sample size (n), mean, SEM, p value, statistical methods and results are presented in *Figure 6D, G and I*.
DOI: https://doi.org/10.7554/eLife.49180.021

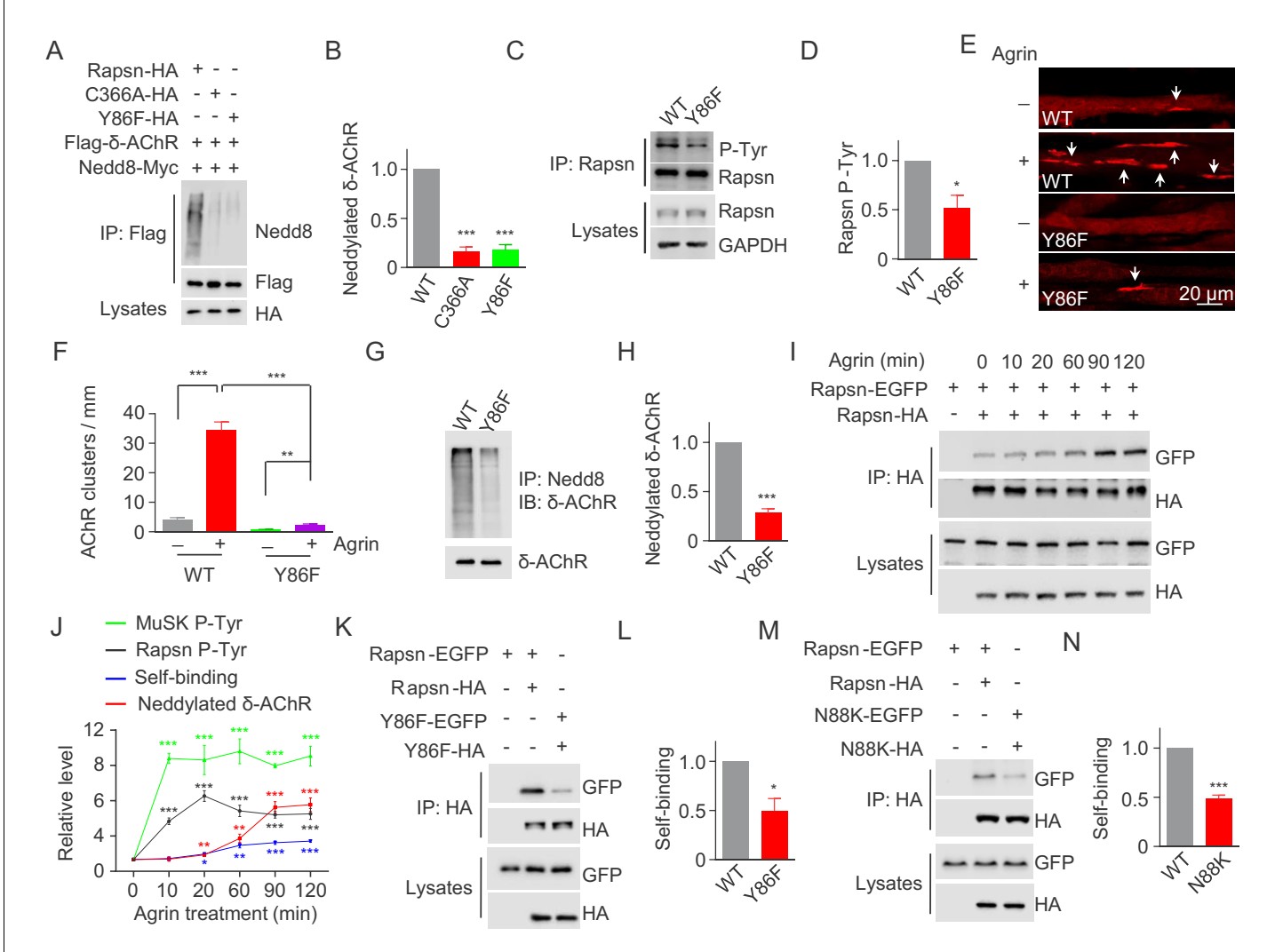

**Figure 7.** Critical roles of Y86 phosphorylation in activating Rapsn E3 ligase activity and AChR clustering. (A) Reduced E3 ligase activity of Rapsn by Y86F mutation in transfected HEK293T cells. (B) Quantitative data of neddylated δ-AChR in (A) (mean ± SEM), ***, p<0.001, One-way ANOVA, n = 3. (C) Reduced Rapsn tyrosine phosphorylation in Y86F mt myotubes, compared with WT controls. (D) Quantitative data of tyrosine phosphorylation of Rapsn in (C) (mean ± SEM), *, p<0.05, unpaired t-test, n = 3. (E) Fewer Agrin-induced AChR clusters in Y86F mt C2C12 myotubes, compared with WT controls. Arrows, AChR clusters. (F) Quantitative data in (E), (mean ± SEM), **, p<0.01, ***, p<0.001, Two-way ANOVA, n = 20 cells. (G) Reduced E3 ligase activity of Rapsn in Y86F mt myotubes. (H) Quantitative data in (G) (mean ± SEM), ***, p<0.001, unpaired t-test, n = 3. (I) Agrin treatment increased Rapsn self-association. Cultured C2C12 cells were transfected with HA- and EGFP-tagged Rapsn, respectively. Resulting myotubes were treated with Agrin. Cell lysates were subjected to co-immunoprecipitation with anti-HA beads, and probed with anti-GFP antibody to reveal the association between Rapsn-HA and Rapsn-EGFP. (J) Quantitative analysis of MuSK and Rapsn tyrosine phosphorylation in *Figure 6A*, neddylated δ-AChR in *Figure 6B*, and Rapsn self-association in *Figure 7I*. Data were shown as mean ± SEM, **, p<0.01, ***, p<0.001 (compared with time 0), One-way ANOVA, n = 3. (K–N) Reduced Rapsn self-association by Y86F (K) or N88K (M) mutation revealed by co-immunoprecipitation in HEK293T cells. Quantitative data of WT or Y86F Rapsn self-association(L); WT or N88K Rapsn self-association (M) (mean ± SEM), *, p<0.05; ***, p<0.001, unpaired t-test, n = 3. Also see *Figure 7—figure supplement 1*.

DOI: https://doi.org/10.7554/eLife.49180.022

The following source data and figure supplements are available for figure 7:

**Source data 1.** Raw data, sample size (n), mean, SEM, p value, statistical methods and results are presented in *Figure 7B, D, F, H, J, L and N*.
DOI: https://doi.org/10.7554/eLife.49180.025

**Figure supplement 1.** Comparable *Rapsn* mRNA level and Rapsn protein stability between WT and Y86F mt myotubes.
DOI: https://doi.org/10.7554/eLife.49180.023

**Figure supplement 1—source data 1.** Raw data, sample size (n), mean, SEM, p value, statistical methods and results are presented in *Figure 7—figure supplement 1C and E*.

*Figure 7 continued*

DOI: https://doi.org/10.7554/eLife.49180.024

was reduced compared with WT muscles. These results demonstrate that in vivo tyrosine phosphory-lated of Rapsn was reduced in N88K mt mice.

To determine whether tyrosine phosphorylation is critical for Rapsn's function, we sought to iden-tify the tyrosine residues that are phosphorylated upon MuSK activation. HA-tagged Rapsn was co-transfected with Flag-tagged MuSK in HEK293T cells, which induced tyrosine phosphorylation of Rapsn (*Figure 6H*), consistent with previous work (*Lee et al., 2008*). Tyrosine phosphorylation was reduced in N88K mt Rapsn (*Figure 6H and I*), as observed in aforementioned studies with C2C12 myotubes (*Figure 6C and D*). We focused on tyrosine residues immediately flanking N88: Y59 in TPR2, Y86 and Y105 in TPR3, and mutated them individually to phenylalanine (F). Tyrosine phosphor-ylation of Y59F and Y105F Rapsn was similar to WT Rapsn in HEK293T cells co-expressing MuSK (*Figure 6H and I*), suggesting that these two residues may not be the target residue. However, Y86F mutation dramatically reduced tyrosine phosphorylation of Rapsn (*Figure 6H and I*). To test this notion further, we generated a recombinant protein containing only TPR domain. As shown in *Figure 6J*, MuSK-mediated tyrosine phosphorylation was completely abolished by the Y86F as well as N88K mutation. Together, these results demonstrate that Y86 may be a site in Rapsn that becomes tyrosine phosphorylated upon MuSK activation, and Y86 phosphorylation is regulated by N88K mutation.

## Y86 phosphorylation for E3 ligase activity and AChR clustering

To investigate the role of Y86 phosphorylation, we determined whether its mutation impacts on E3 ligase activity of Rapsn. HEK293T cells were transfected with Flag-δ-AChR, Myc-tagged Nedd8 and HA-tagged WT, Y86F, or C366A Rapsn and tested δ-AChR neddylation as described in *Figure 5C*. δ-AChR neddylation was reduced in cells expressing Y86F or C366A (*Figure 7A and B*), compared with WT control, suggesting that Y86F mutation impairs the E3 ligase activity of Rapsn. To examine the role of Y86 phosphorylation on AChR cluster formation, we generated Y86F mt C2C12 cells by CRISPR-Cas9 (*Figure 7—figure supplement 1A*). The mutation was confirmed by genomic DNA sequencing (*Figure 7—figure supplement 1B*). Y86F mutation did not alter mRNA or protein levels of Rapsn (*Figure 7—figure supplement 1C*, and *Figure 7C*) or Rapsn protein stability (*Figure 7—figure supplement 1D and E*). Agrin-induced tyrosine phosphorylation was reduced, but not abol-ished in Y86F mt C2C12 myotubes (*Figure 7C and D*), consistent with results of HEK293T cells (*Figure 6H*). Remarkably, Agrin-induced clusters were fewer in Y86F mt C2C12 myotubes, compared with WT control (*Figure 7E and F*), indicating a necessary role of Y86 phosphorylation in Agrin-medi-ated AChR clustering. In support of this notion, δ-AChR neddylation was reduced in Y86F mt C2C12 myotubes, compared with WT control (*Figure 7G and H*).

Next, we investigated how Y86F mutation impairs E3 ligase activity of Rapsn. E3 ligases including those containing RING-domain are regulated by self-association (*Bian et al., 2017*; *Ho et al., 2015*; *Koliopoulos et al., 2016*; *Liew et al., 2010*; *Metzger et al., 2014*; ; *Nikolay et al., 2004*). The TPR domains in the N-terminus of Rapsn are thought to mediate self-association and thus form aggre-gates in heterologous cells (*Maimone and Merlie, 1993*; *Ramarao et al., 2001*). We posited that tyrosine phosphorylation of Rapsn may promote self-association and thus activates E3 ligase activity. To test this, C2C12 myoblasts were co-transfected with Rapsn that were tagged by EGFP and HA, respectively (*Figure 7I*). Resulting myotubes were treated with Agrin and examined for self-binding by co-precipitation. Agrin increased the amount of Rapsn-EGFP co-precipitated by Rapsn-HA, plateaued ~90 min of Agrin treatment. As summarized in *Figure 7J*, upon Agrin stimulation, tyrosine phosphorylation of MuSK occurs prior to that of Rapsn; and subsequently, Rapsn self-association and δ-AChR neddylation were increased, suggesting E3 ligase activity might be regulated by self-association. To examine this, we investigated whether Y86F alters Rapsn self-association. HEK293T cells were transfected with EGFP-tagged and HA-tagged Rapsn. Lysates were incubated with beads immobilized with anti-HA antibody to purify HA-tagged Rapsn and probed with anti-GFP antibody. Compared with WT control, less amount of EGFP-tagged Y86F mt Rapsn was co-precipitated by HA-tagged Y86F Rapsn (*Figure 7K and L*), suggesting impaired ability of mt Rapsn to self-associate.

In agreement, self-association of N88K Rapsn was also reduced, compared with WT Rapsn (*Figure 7M and N*), suggesting that Y86 and N88 are necessary for Rapsn self-association. Together, these results support a working model that MuSK stimulates AChR cluster formation by increasing Rapsn phosphorylation and self-association and thus enhancing its E3 ligase activity.

### Failure of Y86F-Rapsn in rescuing NMJ deficits in N88K mt mice

To investigate the role of Y86 phosphorylation in vivo, we generated adeno-associated virus (AAV, AAV-PHP.B) expressing WT Rapsn (AAV-WT-EGFP) or Y86F mt Rapsn (AAV-Y86F-EGFP) (*Figure 8—figure supplement 1A and B*). Equal volume (10 μl) of AAV viruses in same titer ($1.3 \times 10^{13}$ vg/ml) were intramuscularly injected into N88K mt thigh muscles of embryos at E13. Thigh muscles were isolated at P0 and subjected to morphological characterization. As shown in *Figure 8—figure supplement 1C*, muscles infected with AAV were visible for EGFP. AChR clusters in N88K thigh muscles were reduced in number and appeared to be thin, compared with WT controls (*Figure 8A*, white arrows), as observed in diaphragm muscles (*Figure 1C and D*). Remarkably, AChR clusters were readily detectable in muscles of N88K mt mice that were injected with AAV-WT-EGFP (*Figure 8A*).

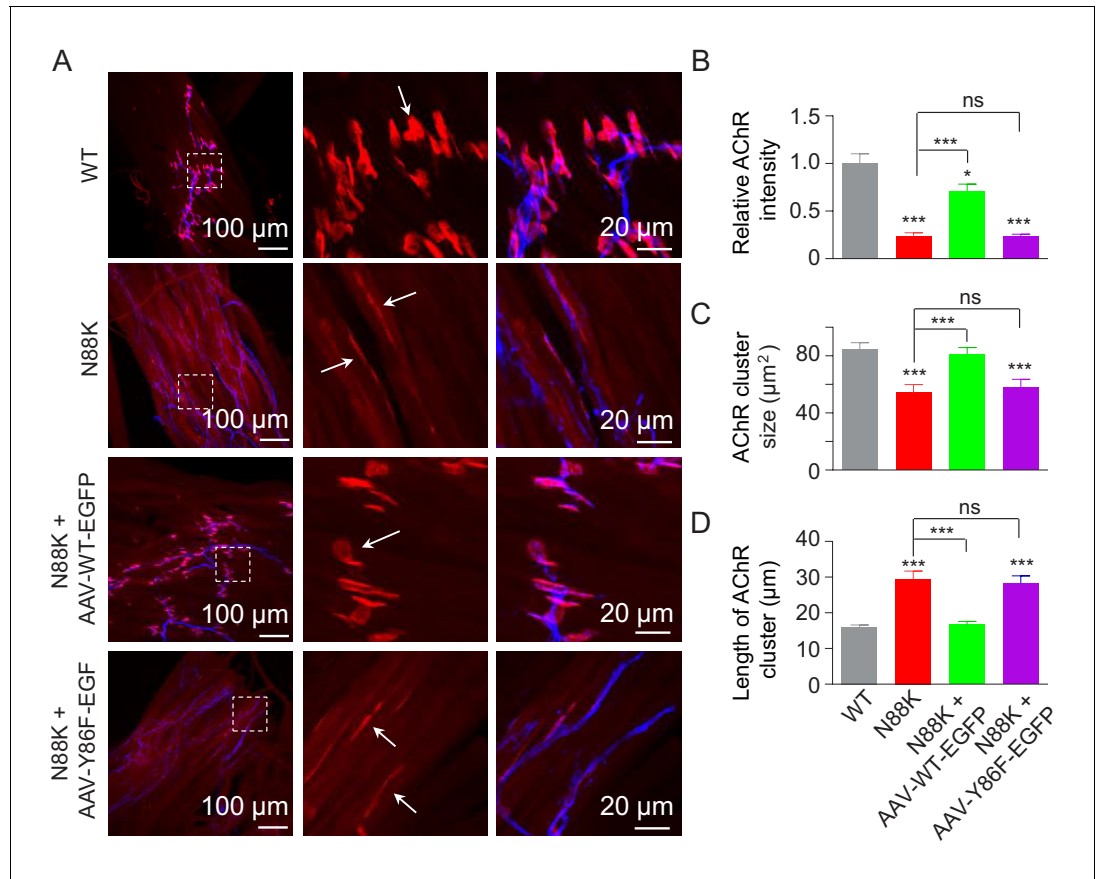

**Figure 8.** Rescue of NMJ deficits by WT Rapsn but not phospho-preventing Y86F mt. (**A**) Thigh muscles of N88K mt were infected with AAV-WT-Rapsn-EGFP (N88K + AAV-WT-EGFP), or infected with AAV-Y86F-Rapsn-EGFP (N88K + AAV-Y86F-EGFP) at E13. The resulting P0 muscles were stained with Flour 594-α-BTX (red) and anti-NF/Syn antibodies (blue) to examine AChR clusters, using uninfected WT or N88K mt as positive or negative controls. (**B–D**) Quantitative analysis of AChR cluster fluorescence intensity (**B**), AChR cluster size (**C**), AChR cluster length (**D**). White arrows, AChR clusters. Data were shown as mean ± SEM; *, $p<0.05$; ***, $p<0.001$, ns, no significant difference, One-way ANOVA, n = 3. Also see *Figure 8—figure supplement 1*.
DOI: https://doi.org/10.7554/eLife.49180.026

The following source data and figure supplement are available for figure 8:

**Source data 1.** Sample size (n), mean, SEM, p value, statistical methods and results are presented in *Figure 8B, C and D*.
DOI: https://doi.org/10.7554/eLife.49180.028
**Figure supplement 1.** Generation of AAV vectors expressing WT or Y86F mt Rapsn and examination of AAV expression.
DOI: https://doi.org/10.7554/eLife.49180.027

Unlike AChR clusters in N88K mt mice that were elongated, oval-shaped clusters were detectable in AAV-WT-EGFP-infected N88K mt thigh muscles. Also increased were the size and fluorescence intensity of AChR clusters (*Figure 8A–D*). These results demonstrate that N88K mutation-caused NMJ impairment could be attenuated by viral expression of WT Rapsn. Noticeably, the rescue effects were not observed in mice injected with AAV-Y86F-EGFP (*Figure 8A–D*). These results provide in vivo evidence that Y86 is critical to Rapsn's function in NMJ formation.

## Discussion

This study provides evidence that the N88K mutation prevented muscle fibers from forming aneural AChR clusters prior to innervation as well as nerve-induced AChR clusters. Axon terminals were extensively arborized in N88K mt mice, with increased secondary and tertiary nerve branches and reduced density of synaptic vesicles phenotypes observed in mt mice lacking LRP4 or MuSK (*DeChiara et al., 1996*; *Weatherbee et al., 2006*; *Wu et al., 2012b*). These results uncover previously unappreciated impacts of N88K mutation on the NMJ and contribute to a better understanding of pathological mechanisms of how N88K mutation causes CMS. At the molecular level, studies of N88K mutation led to discover a pathway by which signal is transduced from MuSK to Rapsn. Agrin is a factor utilized by motoneurons to direct NMJ formation, in particular postsynaptic differentiation (*McMahan, 1990*). It binds to LRP4 and thus activates MuSK (*Kim et al., 2008*; *Zhang et al., 2008*; *Zong et al., 2012*). Like Rapsn, Agrin, LRP4, and MuSK are absolutely necessary for NMJ formation (*DeChiara et al., 1996*; *Gautam et al., 1996*; *Gautam et al., 1995*; *Weatherbee et al., 2006*). However, signaling events downstream of MuSK were unclear except that Dok7 is a necessary adapter protein that is believed to dimerize MuSK (*Okada et al., 2006*); and Rapsn that could function as an adapter protein to bridge AChRs to the cytoskeleton and/or as an E3 ligase to promote neddylation of the AChR (*Li et al., 2016*; *Li et al., 2018*). Although tyrosine phosphorylation of Rapsn has been observed in torpedo electric organs (*Mohamed and Swope, 1999*) and in MuSK- and Rapsn-cotransfected cells (*Lee et al., 2008*), its functional significance was unclear. The N88K mutation seemed to have little effect on the levels of Rapsn or AChR in muscle cells. We demonstrate that it may alter the phosphorylation of Y86, an adjacent tyrosine residue that is required for Agrin signaling. First, Y86 became tyrosine phosphorylated upon Agrin stimulation. Second, mutating Y86 to F, to block tyrosine phosphorylation, reduced MuSK-dependent tyrosine phosphorylation of full-length Rapsn, and abolished MuSK-dependent tyrosine phosphorylation of TPR1-7 of Rapsn, suggesting Y86 is a major tyrosine-phosphorylation site at TPR domain. Third, in HEK293T cells, Y86F mutation reduced Rapsn E3 ligase activity to comparable, minimal level as Rapsn C366A mt whose E3 ligase activity was abolished (*Li et al., 2016*). In agreement, E3 ligase activity of Rapsn was reduced in Y86F knock-in myotubes, suggesting a necessary role of Y86 phosphorylation in E3 ligase activity regulation. Fourth, WT Rapsn was able to rescue AChR clustering deficits in N88K mt mice; this rescue effect was not observed with Y86F mt Rapsn. Finally, we demonstrate that N88 and Y86 are both necessary for Rapsn self-association, a process that has been shown to be necessary for E3 ligase activation (*Bian et al., 2017*; *Ho et al., 2015*; *Koliopoulos et al., 2016*; *Liew et al., 2010*; *Metzger et al., 2014*; *Nikolay et al., 2004*; ). A parsimonious explanation of these results is that MuSK activation causes, directly or indirectly, Rapsn tyrosine phosphorylation at Y86, which promotes Rapsn self-association and thus activate the E3 ligase activity, revealing a mechanism for Rapsn activation. These results also suggest a novel, molecular pathological mechanism by which N88K mutation impairs the NMJ formation and maintenance.

Rapsn has 14 tyrosine residues, including 12 in the TPR domain. Mutation of the Y86 abolished MuSK-dependent tyrosine phosphorylation of the TPR domain, suggesting that Y86 is a key tyrosine phosphorylation site in this region or its mutation prevents others tyrosine residues from being phosphorylated. Interestingly, tyrosine phosphorylation of WT Rapsn was reduced, but not abolished by the Y86F mutation. About 40% of MuSK-dependent tyrosine phosphorylation remained in Y86F mt. These results could suggest that tyrosine residues in C-terminal regions of Rapsn may be a target for MuSK regulation. This region contains a coiled-coil domain that could bind to the AChR (*Ramarao and Cohen, 1998*), and a RING-H2 domain that could bind to β-dystroglycan (*Bartoli et al., 2001*) and possess the E3 ligase activity (*Li et al., 2016*). It would be interesting to determine whether these activities are regulated by tyrosine phosphorylation. Consistent with this

notion, Agrin treatment can enhance the AChR-Rapsn interaction (*Luo et al., 2008*; *Moransard et al., 2003*) and the association of the AChR-Rapsn complex to cytoskeleton (*Moransard et al., 2003*).

Both MuSK kinase activity and Rapsn are necessary for prepatterning or the formation of aneural AChR clusters (*Lin et al., 2001*; *Yang et al., 2001b*); and Rapsn is known to interact with MuSK in the absence of Agrin (*Apel et al., 1997*). Further, LRP4 could interact with MuSK and thus maintain its activity in the absence of Agrin (*Kim et al., 2008*; *Zhang et al., 2008*). Indeed, Rapsn phosphorylation could be detectable at E14, when aneural AChR clusters are yet to be innervated by axons. Together, these observations suggest that tyrosine phosphorylation of Rapsn could contribute to the formation of aneural AChR clusters or muscle prepatterning. It is worthy pointing out that evidence is lacking whether Rapsn is a direct substrate protein of MuSK.

Pathological mechanisms of the N88K mutation have been examined previously. It was shown to reduce Rapsn's ability to induce AChR clusters in HEK and TE671 cells (*Cossins et al., 2006*; *Ohno et al., 2002*). In agreement, the N88K mt was shown here to reduce Rapsn's ability to induce AChR clusters in HEK293T cells (*Figure 4A*). In N88K knock-in mt mice and in CRISPR/Cas9-generated C2C12 myotubes, AChR clusters were almost diminished. However, N88K seemed to be able to induce AChR clusters in *Rapsn* mt myotubes (*Cossins et al., 2006*). The cause of this discrepancy is unclear. In the current study, the impact of the N88K mutation was inferred mainly by loss-of-function approaches, both in vitro and in vivo, without changing endogenous Rapsn level (*Figure 1* and *Figure 4E and F*). Rapsn is known to induce AChR clusters at low concentrations, but inhibit AChR clusters at high level (*Han et al., 1999*; *Yoshihara and Hall, 1993*). This inverted U-function could complicate the interpretation of results of overexpressing WT or mt Rapsn.

CMS patients carrying the N88K mutation show variable phenotypes, from serious, early-onset symptoms to mild, late-onset symptoms (*Cossins et al., 2006*; *Milone et al., 2009*; *Müller et al., 2003*), even in patients with homozygous N88K mutation. In severe cases, N88K heteroallelic with c.966 + 1GT, L14P, or a frameshift mutation may cause postnatal death (*Maselli et al., 2003*; *Milone et al., 2009*; *Richard et al., 2003*), indicating N88K mutation could have seriously detrimental effects on NMJ formation or maintenance in human subjects. Overall, the symptoms of CMS patients with homozygous N88K mutation are milder than that of patients carrying heteroallelic N88K mutation with a second mutation. These secondary mutations including sequence deletions or insertions, such as c.1177-1178del AA, p.V50-S55del, p.K373del, and c.553-554insGTTCT, are often more disruptive (*Milone et al., 2009*). We also note that phenotypes of N88K mt mice seem to be more severe than symptoms of CMS patients. Almost all N88K mt mice died within 24 hr of birth. A simple explanation of these observations may be genetic difference. Gene mutations identified in patients with various diseases such as Lesch-Nyhan syndrome, Lowe syndrome, galactosemia diseases often failed to reproduce clinic symptoms in mouse models (*Elsea and Lucas, 2002*). On the other hand, a knock-in mutation of AQP2, a gene implicated in hereditary non-X-linked nephrogenic diabetes insipidus, causes more severe phenotypes in mice than in human subjects (*Yang et al., 2001a*). Even in same species, effects of a mutation could have variable impacts on different strains of mice, for example, the Agrin mutation, Agrin[nm380] (*Bogdanik and Burgess, 2011*). Furthermore, it is possible that some patients with the N88K mutation are unable to survive and thus were not included in clinic reports.

In sum, this study demonstrates that the most prevalent CMS-associated mutation in Rapsn is causal to NMJ deficits. Not only does it reveal pathological mechanisms of how the N88K mutation alters NMJ development, the study also uncovers molecular mechanisms by which Rapsn is regulated upon MuSK activation. The results support a model where Rapsn becomes phosphorylated on tyrosine residues for self-association and E3 ligase activity, which are necessary for AChR clustering. Intriguingly, NMJ deficits in N88K mt mice could be diminished by a virus expressing WT Rapsn, suggesting that gene therapy may be beneficial for CMS.

# Materials and methods

## Key resources table

| Reagent type (species) or resource | Designation | Source or reference | Identifiers | Additional information |
|---|---|---|---|---|
| Genetic reagent (*M. musculus*) | *Rapsn-/-* | PMID: 7675108 | | |
| Genetic reagent (*M. musculus*) | *N88K* | This paper | | See detail information in Materials and methods . |
| Cell line (*Homo sapiens*) | HEK293T | ATCC | Cat#:CRL-3216 RRID: CVCL_0042 | From ATCC; Cell identity has been confirmed by STR profiling and cell line was found to be free of *Mycoplasma*. |
| Cell line (*M. musculus*) | C2C12 | ATCC | Cat#: CRL-1772 RRID: CVCL_0188 | From ATCC; Cell identity has been confirmed by STR profiling and cell line was found to be free of *Mycoplasma*. |
| Cell line (*M. musculus*) | N88K mt C2C12 | This paper | | See detail information in Material and methods ). |
| Cell line (*M. musculus*) | Y86F mt C2C12 | This paper | | See detail information in Material and methods . |
| Cell line (*M. musculus*) | *Rapsn-/-* C2C12, clone 11–7 | PMID: 10414969 | | |
| Antibody | Mouse monoclonal anti-HA Agarose | Thermo Fisher Scientific | Cat#: 26181, RRID: AB_2537081 | 1: 40 for IP |
| Antibody | Goat anti-rabbit IgG conjugated with Alexa Fluor 488 | Thermo Fisher Scientific | Cat#: A-11008 RRID: AB_10563748 | IHC (1:500) |
| Antibody | Rabbit polyclonal anti-Actin | Cell Signaling Technology | Cat #: 4967, RRID: AB_330288 | WB (1:10000) |
| Antibody | Rabbit polyclonal anti- α-Actin | Abcam | Cat #: ab52218 RRID: AB_870573 | WB (1:1000) |
| Antibody | Rabbit polyclonal anti-Flag | Sigma-Aldrich | Cat #: F7425 RRID: AB_439687 | WB (1:1000) |
| Antibody | Mouse monoclonal anti-GAPDH (Clone, 6C5) | Santa Cruz Biotechnology | Cat #: sc-32233, RRID: AB_627679 | WB (1:10000) |
| Antibody | Rabbit polyclonal anti-neurofilament | Cell Signaling Technology | Cat#: 2837, RRID: AB_823575 | IHC (1:1000) |
| Antibody | Rabbit polyclonal anti-synapsin | Cell Signaling Technology | Cat#: 5297, RRID: AB_2616578 | IHC (1:1000) |
| Antibody | Mouse monoclonal anti-Ubiquitin (Clone, Ubi-1) | Abcam | Cat#: ab7254, RRID: AB_305802 | WB (1:1000) |
| Antibody | Rabbit polyclonal anti-GFP | Cell Signaling Technology | Cat#: 2555, RRID: AB_10692764 | WB (1:1000) |
| Antibody | Mouse monoclonal P-Tyr-100 | Cell Signaling Technology | Cat #: 9411 RRID: AB_331228 | WB (1:1000) |

*Continued on next page*

*Continued*

| Reagent type (species) or resource | Designation | Source or reference | Identifiers | Additional information |
|---|---|---|---|---|
| Antibody | Mouse monoclonal Anti-Flag affinity gel (Clone, M2) | Sigma-Aldrich | Cat#: A2220, RRID: AB_10063035 | 1: 50 for IP |
| Antibody | Rabbit polyclonal anti-HA | Sigma-Aldrich | Cat#: H6908, RRID: AB_260070 | WB (1:2000) |
| Antibody | Mouse monoclonal anti-Rapsn (Clone 1234) | Abcam | Cat#: ab11423, RRID: AB_298028 | WB (1:1000) |
| Antibody | Rabbit polyclonal anti-Transferrin | Abcam | Cat#: ab82411, RRID: AB_1659060 | WB (1:1000) |
| Antibody | Mouse monoclonal anti-δ-AChR (Clone, 88B) | Thermo Fisher Scientific | Cat#: MA3-043, RRID: AB_2081037 | WB (1:1000) |
| Antibody | Rabbit polyclonal anti-Rapsn | PMID: 18940591 | | WB (1:1000) |
| Antibody | Goat polyclonal anti-α-AChR | PMID: 3484485 | | WB (1:1000) |
| Antibody | Rabbit polyclonal anti-β-AChR | PMID: 3484485 | | WB (1:1000) |
| Antibody | Horseradish peroxidase (HRP)-conjugated goat anti-rabbit IgG | Thermo Fisher Scientific | Cat#: 32260, RRID: AB_1965959 | WB (1:5000) |
| Antibody | Horseradish peroxidase (HRP)-conjugated goat anti-Mouse IgG | Thermo Fisher Scientific | Cat#: 32230, RRID: AB_1965958 | WB (1:5000) |
| Antibody | Horseradish peroxidase (HRP)-conjugated goat anti-Rat IgG | Thermo Fisher Scientific | Cat#: 31470, RRID: AB_228356 | WB (1:5000) |
| Recombinant Protein | Agrin | R and D Systems | Cat#: 550-AG-100 | 50 ng / ml for induction of AChR clusters in culture myotubes |
| Other | Immunofluorescence of NMJ in diaphragm and muscles | PMID: 18278041 PMID: 22794264 | | |
| Other | Electron microscopic analysis | | | |
| Other | Generation of Gene-modified C2C12 cells | PMID: 27839998 | | |
| Other | Isolation of cell surface protein | PMID: 22157653 | | |
| Other | Isolation of surface AChR and their associated proteins | PMID: 18940591 | | |
| Other | AAV Virus production | PMID: 30626963 and Information in Addgene: https://www.addgene.org/protocols/aav-production-hek293-cells/ | | |

## Mice

*Rapsn* null (-/-) mt mice were a kind gift from Dr. Peter Noakes (*Gautam et al., 1995*). The N88K mt mice were generated by the University of Rochester Mouse Genome Editing Resource using CRISPR/Cas9 approach, as described previously (*Li et al., 2016*). Briefly, a single strand DNA template (ssDNA), CCGTG GTCCA GATTG ATACT GCTCG GGGAC TGGAG GATGC TGACT TCCTG

CTCGA AAGCT ACCTC AAGTT AGCTC GCAGC AATGA GAAGC TATGT GAGTT CCACA AAACC ATCTC CTACT GCAAG ACCTG CCTCG G, containing the desired mutations was synthesized and purified by Integrated DNA Technologies. SgRNA was generated by a MEGAshortscript T7 kit (Life Technologies) using template sequence (GCTCG AAAGC TACCT GAACC) cloned in pX330 plasmid (Addgene #42230) and was purified using MEGAclear kit (Life Technologies). Mixture of Cas9 mRNA (TriLink Biotechnologies, 100 ng / μl), sgRNA (50 ng / μl), and ssDNA (100 ng / μl) was injected into fertilized eggs from C57BL/6J mice (Jackson Laboratory, Stock #000664). Viable two-cell stage embryos were transferred to pseudo-pregnant ICR females to generate founder mice, which were subsequently bred with C57BL/6J mice for germline transmission to generate F1 mice. Mice carrying N88K mutation were screened by PCR analysis and confirmed by DNA sequencing. Primers: 5'-GCTCG AAAGC TACCTG AACCT GGCG-3' and 5'-CACGA GGTTCT CAGGG AGCCT CA-3' were used to verify WT genomic DNA, and primers: 5'-CTTCC TGCTC GAAAG CTACC TCAAG TTAGC T-3' and 5'-CACGA GGTTC TCAGG GAGCC TCA-3' were used to verify N88K mt genomic DNA. C57BL/6 mice were used as WT controls. Mice were housed in cages in a room with 12 hr light-dark cycle with ad libitum access to water and rodent chow diet (Diet P3000). Embryo and P0 pups of either sex were analyzed, unless otherwise indicated. Animal protocols have been approved by the Institutional Animal Care and Use Committee of Case Western Reserve University.

## Reagents and antibodies

Following reagents were purchased from Thermo Fisher Scientific (Waltham, MA): Flour-594 conjugated α-Bungarotoxin (Flour 594-α-BTX; Catalog number, B-13423; 1:1000 for staining), Biotin conjugated α-Bungarotoxin (Biotin-α-BTX, Catalog number, B1196), Sulfo-NHS-SS-Biotin (Catalog number, PG82077), streptavidin-coupled agarose beads (Catalog number, 20349), goat anti-rabbit IgG conjugated with Alexa Fluor 488 (Catalog number, A-11008; 1: 500 for staining), mouse anti-HA agarose (Catalog number, 26181). Mouse anti-GAPDH (Catalog number, NB 600–501; 1:10000 for WB) was from Novus (Littleton, CA). Rabbit anti-Actin (Catalog number, 4967; 1: 5000 for western blotting), rabbit anti-Nedd8 (Catalog number, 10695299; 1:1000 for WB and IP), rabbit anti-neurofilament (Catalog number, 2837S; C28E10, 1:1000 for staining), rabbit anti-synapsin (Catalog number, 5297; D12G5, 1:1000 for staining) and mouse anti-phospho-tyrosine (Catalog number, 9411S; P-Tyr-100; 1:1000 for western blotting) were from Cell signaling technology (Boston, MA). Cycloheximide (CHX, Catalog number, C7658-5G; 50 μg / ml), rabbit anti-HA (Catalog number, H6908; 1:1000 for western blotting), rabbit anti-Flag (Catalog number, F7425; 1:1000 for western blotting), HA peptide (Catalog number, 11666975001), anti-Flag M2 affinity gel (Catalog number, A2220) were from Sigma (Mendota Heights, MN). Mouse anti-Ubiquitin (Catalog number, ab7254; 1:1000 for western blotting), rabbit anti-α-Actin (Catalog number, ab52218; 1: 10000 for western blotting), mouse anti-Rapsn (Catalog number, ab11423; clone 1234, 1:1000 for western blotting), rabbit anti-Transferrin (Catalog number, ab82411; 1: 1000 for western blotting) were from Abcam (Cambridge, MA). Mouse anti-δ-AChR (C4; 1:1000 for western blotting) was from Santa Cruz Biotechnology (Dallas, TX). Rabbit anti-Rapsn (2741), goat anti-α-AChR and rabbit anti-β-AChR antibodies were described previously (*Barik et al., 2014*; *Luo et al., 2008*; *Wu et al., 2012a*; *Zhao et al., 2017*). Horseradish peroxidase (HRP)-conjugated goat anti-rabbit IgG (Catalog number, 32260), goat anti-mouse IgG (Catalog number, 32230), goat anti-rat IgG (Catalog number, 31470) antibodies (1:5000 for western blotting) were from Pierce (Rockford, IL).

## Immunofluorescence

The procedures for staining muscles or diaphragms were described previously (*Li et al., 2016*; *Li et al., 2008*; *Wu et al., 2012b*). Briefly, skeletal muscles or diaphragms were fixed in 4% paraformaldehyde in PBS for 24 hr, and then dissected in PBS. Dissected samples were rinsed with 0.1 M glycine in phosphate-buffered saline (PBS) for 1 hr at room temperature and followed by three-time washing in PBS. Samples were then incubated with the blocking buffer (5% BSA, 5% goat serum, 1% Triton X-100 in PBS) for 1 hr at room temperature and incubated with primary antibodies in blocking buffer at 4°C overnight. Next day, after washing three times with washing buffer (1% Triton X-100 in PBS), samples were incubated with fluorescent-labeled secondary antibodies at room temperature for 1–2 hr. Samples were then washed with washing buffer and mounted with Vectashield mounting medium (H1000, Vector Laboratories, Burlingame, CA) and coverslip. Images were collected with a

Zeiss confocal laser scanning microscope (LSM 800) and collapsed into a single image. AChR clusters and axon branches in left, ventral diaphragms were quantified within 1 mm of primary phrenic nerve branches.

## Western blot analysis

Western blot was performed as described previously (*Barik et al., 2014*; *Zhao et al., 2017*).

## Electron microscopic analysis

The procedures for electron microscopic were described previously (*Li et al., 2016*; *Wu et al., 2015*; *Wu et al., 2012b*).

## Electrophysiological recording

The procedures for electrophysiological recording of P0 diaphragms were described as previously (*Li et al., 2008*; *Shen et al., 2018*).

## C2C12 culture and generation of Gene-modified C2C12 cells

Mouse myoblasts C2C12 were purchased from ATCC (Catalog number, CRL-1772, Manassas, VA). *Rapsn* null mt muscle cells (clone 11–7) were kindly provided by Dr. C. Fuhrer. The procedures for C2C12 culture and generation of knock-in mt C2C12 cell by CRISPR-Cas9 were described previously (*Li et al., 2016*; *Zhang et al., 2012*; *Zhang et al., 2007*). The sgRNA carrying targeting sequence GCTCG AAAGC TACCT GAACC was used for generation of N88K mt C2C12; sgRNA carrying targeting sequence CAT TGC TGC GCG CCA GGT TC was used for generation of Y86F mt C2C12. For comparing Rapsn tyrosine phosphorylation and E3 ligase activity between WT and N88K mt culture myotubes, myotubes were treated with Agrin for 2 hr, because Agrin activates MuSK within 10 min of stimulation and tyrosine phosphorylation of Rapsn that plateaus around 20 min (*Figure 6A*), and Agrin induces the formation of AChR clusters beginning around 2 hr of stimulation (*Ngo et al., 2012*). For studying Rapsn self-association in culture myotubes (*Figure 7I*), overexpression of high dosage of Rapsn could induce Rapsn aggregation in the absence of Agrin (data not shown), several dosages of Rapsn vectors were tested, and 3.5 µg Rapsn-HA vector and 3.5 µg Rapsn-EGFP vector were used to transfect 10 cm culture plate. At these dosages, through viewing GFP fluorescence, we could see expression of Rapsn at culture myotubes, but these myotubes didn't have obvious Rapsn aggregates.

## Isolation of cell surface protein

The procedures for isolating surface proteins were described previously (*Chen et al., 2012*). Briefly, cells were washed twice with ice-cold PBS followed by incubation with 10 ml 0.25 mg/mL Sulfo-NHS-SS-Biotin in PBS containing $Mg^{2+}/Ca^{2+}$ at 4°C for 45 min. Reaction was quenched by adding 10 mM Glycine. Cells were lysed in RIPA buffer containing 150 mM NaCl, 25 mM Tris-HCl, 1% Triton X-100, 10% glycerol, 25 mM NaF, 2 mM NaVO₃, 5 mM $Na_4P_2O_7$, protease Inhibitor cocktail (CO-RO, Roche), PH 7.4 and centrifuged. Biotin-labeled surface proteins in supernatant were precipitated by adding Streptavidin-coupled agarose beads, and dissolved in 2 X protein loading buffer (20% Glycerol, 100 mM Tris-HCl, 4% SDS, 2% β-mercaptoethanol, 0.01% bromophenol blue), and analyzed by SDS-PAGE gel.

## Isolation of surface AChR and their associated proteins

The procedure for isolating surface AChR and associated proteins were described previously (*Luo et al., 2008*). Briefly, live cultured myotubes were washed with cold PBS twice, and then incubated with 300 nM biotin-α-BTX in PBS containing $Mg^{2+}/Ca^{2+}$ at 4°C for 1 hr. After washing with PBS for three times, cells were lysed in RIPA buffer. Lysates were centrifuged and supernatants were incubated with streptavidin beads for 6 hr at 4°C to precipitate biotin-labeled protein. Precipitated proteins were resolved by 2 X protein loading buffer and analyzed by SDS-PAGE gel.

## Virus production

The procedures for generating recombinant AAV virus were as described previously (*Challis et al., 2019*). Briefly, full length WT or Y86F mt Rapsn was subcloned into AAV vector (AAV-CMV-GFP;

Addgene Catalog, 67634). AAV vectors expressing WT Rapsn or Y86F mt Rapsn were cotransfected with helper vector (From Agilent), capsid vector (pUCmini-iCAP-PHP.B; Addgene Catalog, 103002) were cotransfected into human embryonic kidney (HEK) 293 T cells. Medium was harvested twin at 72 hr and 120 hr after transfection and viruses in the medium were precipitated by 40% PEG 8000. Cells were finally harvested and lysed in SAN digestion buffer (0.5 M NaCl, 4 mM Tris, 1 mM $MgCl_2$). Virus was purified by using a discontinuous iodixanol gradient. The titer was determined by qPCR.

## Statistical analysis

Data were analyzed by unpaired t-test, One-way ANOVA and Two-way ANOVA. Unless otherwise indicated, data were shown as mean ± SEM. Statistical difference was considered when $p < 0.05$.

## Acknowledgements

We thank Dr. Peter Noakes (The University of Queensland, School of Biomedical Science) for providing *Rapsn* null mutants; Dr. Lin Gan (Rochester Mouse Genome Editing Resource Core) for generating N88K mt mice; members of Mei/Xiong Lab for critical comments. This study was supported in part by grants from NIH and Veterans Affairs to LM and WCX.

## Additional information

### Funding

| Funder | Grant reference number | Author |
|---|---|---|
| National Institutes of Health | NS082007 | Lin Mei |
| National Institutes of Health | NS090083 | Lin Mei |
| National Institutes of Health | AG051510 | Lin Mei |
| Veterans Health Administration office of Research and Development | 1/01IB001020A | Lin Mei |
| National Institutes of Health | AG45781 | Wen-Cheng Xiong |
| National Institutes of Health | AG060997 | Wen-Cheng Xiong |

The funders had no role in study design, data collection and interpretation, or the decision to submit the work for publication.

### Author contributions

Guanglin Xing, Conceptualization, Resources, Data curation, Software, Formal analysis, Validation, Methodology, Writing—original draft, Writing—review and editing; Hongyang Jing, Conceptualization, Resources, Data curation, Formal analysis, Validation, Methodology; Lei Zhang, Data curation, Software, Formal analysis, Methodology; Yu Cao, Data curation, Software, Formal analysis, Investigation, Methodology; Lei Li, Data curation, Formal analysis; Kai Zhao, Data curation, Software, Methodology; Zhaoqi Dong, Wenbing Chen, Data curation, Software; Hongsheng Wang, Rangjuan Cao, Data curation; Wen-Cheng Xiong, Conceptualization, Project administration; Lin Mei, Conceptualization, Resources, Data curation, Supervision, Funding acquisition, Methodology, Writing—original draft, Project administration, Writing—review and editing

### Author ORCIDs

Guanglin Xing ⑩ https://orcid.org/0000-0002-8258-0293
Hongyang Jing ⑩ https://orcid.org/0000-0003-2811-1266
Yu Cao ⑩ http://orcid.org/0000-0003-3708-575X
Lin Mei ⑩ https://orcid.org/0000-0001-5772-1229

## Ethics

Animal experimentation: This study was performed in strict accordance with the recommendations in the Guide for the Care and Use of Laboratory Animals of the National Institutes of Health. Animals were handled according to the approved institutional animal care and use committee (IACUC) protocol (2017-0115) of Case Western Reserve University.

## Decision letter and Author response

Decision letter https://doi.org/10.7554/eLife.49180.031
Author response https://doi.org/10.7554/eLife.49180.032

## Additional files

### Supplementary files

• Transparent reporting form
DOI: https://doi.org/10.7554/eLife.49180.029

### Data availability

All data generated or analysed during this study are included in the manuscript and supporting files. Source data about statistical results are provided in Excel files, including Figures 1E-J, 2E-H, 2M, 2N, 3A, 3B, 3D, 3E, 4B, 4F, 5D, 5F, 5H, 6D, 6G, 6I, 7B, 7F, 7H, 7J, 7L, 7N, 8B, 8C, 8D; Figure 1–figure supplement 1C, 1E, 1H, Figure 4–figure supplement 1B, Figure 4–figure supplement 2B and 2D, Figure 5–figure supplement 1D, Figure 7–figure supplement 1C and E.

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
