## [Decision Letter]

Thank you for submitting your article "A novel mechanism in agrin signaling revealed by a prevalent rapsyn mutation in congenital myasthenic syndrome" for consideration by *eLife*. Your article has been reviewed by Gary Westbrook as the Senior Editor, a Reviewing Editor, and three reviewers. The following individuals involved in review of your submission have agreed to reveal their identity: Hans Brenner (Reviewer #1); Markus Ruegg (Reviewer #3). The reviewers have discussed the reviews with one another and the Reviewing Editor has drafted this decision to help you prepare a revised submission.

Summary:

This manuscript examined a knockin mouse carrying a mutation (N88K) of rapsyn, a common mutation in congenital myasthenic syndrome. The homozygous KI mice died shortly after birth with NMJ deficits. The authors implicate the agrin-LRP4-MuSK tyrosine phosphorylation of rapsyn in the pathophysiology. The authors recently reported that rapsyn has E3 ligase activity, which is essentially for its role in AChR clustering. The importance of this signaling cascade is known and many of the general observations regarding rapsyn are known, the new information in this manuscript is the link of the N88K mutation to impaired rapsyn phosphorylation at Y86. The experiments appear carefully done and described and the current results are a logical extension of the authors' prior studies and provide additional details on the role of rapsyn at the NMJ.

Essential revisions:

Although the reviewers were supportive of the manuscript, a number of essential issues will need to be addressed in a revised manuscript as listed here and explained in depth in the full comments of the reviewers below. The authors should also edit the manuscript to be careful not to overstate their conclusions in the title or text of the manuscript.

Reviewer 1 – the number of observations in each experiment should be documented.

Reviewer 2 –

1. Some of the data are overinterpreted.

2. Is rapsyn tyrosine phosphorylated at E14?

3. Self-association of N88K rapsyn.

4. Basis of variability of human phenotypes relative to the KI mouse must be explained/addressed.

5. Association with cytoskeletal proteins?

6. Figure 1 – comparison needed of WT, hets and homozygous, ephys?

7. Figure 2 – quality of staining of clusters at E14 is poor.

8. Figure 5C – explain Agrin treatment protocol.

Reviewer 3 – please address the apparent discrepancy with N88K mutations in humans.

*Reviewer #1:*

The neuromuscular junction has for decades been a model synapse for investigating basic mechanisms of synaptic transmission and synapse development. Motor neuron growth cones differentiate into presynaptic nerve terminals at the site of n.-m. contact, and in close register, muscle fibers form high density accumulations of receptors for the transmitter acetylcholine (AChRs). Synaptic AChR clustering is driven by agrin secreted from the motor nerve terminal and activating its effector MuSK in the muscle fiber. AChR clustering ultimately involves the anchoring of AChRs to the synaptic actin cytoskeleton through the organization of a subsynaptic apparatus, a complex of transmembrane and peripheral membrane proteins. Whereas their importance for clustering has been demonstrated genetically, the nature of their interactions, i.e. purely physical and/or involved in signaling, and their respective relevance for AChR clustering have remained largely unknown, although it would be required for full molecular understanding of NMJ development. One of the major components of the subsynaptic apparatus is rapsyn, a peripheral membrane protein linking AChRs to the synaptic actin cytoskeleton via unknown mechanisms. In the absence of rapsyn, AChRs do not cluster, and different mutations in rapsyn gene are linked to potentially life-threatening NMJ malfunctions – a subgroup of NMJ diseases called congenital myasthenic syndromes (CMS).

To dissect novel mechanisms of rapsyn's function, the authors have taken advantage of an earlier finding that one form of CMS is linked to a common mutation in rapsyn, in which asparagine 88 is mutated to lysine (N88K). Specifically, they generated rapsyn N88K mutant mice and observed impairments in NMJ development and function, with reduced synapse number, AChR density, presynaptic vesicle density and reduced synaptic folds; combined this led to cyanosis and caused death within 24 hours after birth. Reminiscent synaptic phenotypes have been reported in CMS patients carrying the N88K mutations.

In molecular and cell biological experiments on heterologous systems and in cultured muscle cells expression of N88K-mutant reduced spontaneous and agrin-induced AChR clustering without an effect on total or surface rapsyn, respectively, demonstrating reduced ability of N88K to cluster AChRs. This was not due to impaired binding of N88K to AChRs.

In previous experiments, the same group found that rapsyn possesses E3 ligase activity that is required for its ability to cluster AChRs, apparently through a process involving posttranslational neddylation of the AChRs. They now show in biochemical experiments both in HEK293 cells, in cultured muscle cells and in mouse muscle carrying the N88K mutation that indeed neddylation was strongly reduced compared to cells/tissue expressing wild type rapsyn. Evidence is then presented that agrin/MuSK phosphorylates a Tyr residue (Y86) in rapsyn immediately flanking N88 in WT, but not in N88K mutant muscle, and that phosphorylation at Y86 is required for rapsyn's E3 ligase activity and for its ability to cluster AChRs.

With activity being regulated by self-association in numerous E3 ligases, the authors then successfully tested their hypothesis that self-association of rapsyn in HEK293 cells – a condition for forming aggregates in heterologous cells – was impaired with non-phosphorylatable rapsyn Y76F mutants, as it was with rapsyn N88K mutants. They validated the physiological relevance of their hypothesis in vivo, by injecting fetal N88K mutant muscles with AAV expressing WT and N88K mutant rapsyns, respectively, which mimicked the changes in AChR clustering in the genetically modified mutant compared to control WT mice. Thus, NMJ deficits in N88K mutant muscle could be rescued by w.t. rapsyn, but not with N88K rapsyn. In contrast, forced expression of Y86F rapsyn did not rescue the synaptic phenotype in N88K mutants.

Combined, the data suggest that activated MuSK phosphorylates rapsyn at Y86 which stimulates rapsyn self-association, thus activating its E3 ligase activity which is required for AChR clustering. The paper provides not only insight into the molecular mechanism agrin-induced AChR clustering at the NMJ, but also explains, at least in part, the synaptic phenotype of rapsyn N88K patients afflicted by this form of CMS.

The data presented are well documented, except that numbers of items examined (AChR clusters, animals) should be given (see e.g. (4) below). Validations by appropriate controls are convincing.

*Reviewer #2:*

Essential revisions:

In the present study, the authors investigated how a prevalent mutation in a well-known congenital myasthenic syndroms (CMS) causing gene (ie Rapsyn) leads to neuromuscular junction (NMJ) formation defects in rodent. Rapsyn is a peripheral membrane protein that binds directly to the skeletal muscle acetylcholine receptors (AChR) and is essential for AChR clustering in the postsynaptic membrane. However, the mechanisms underlying Rapsyn function in anchoring postsynaptic proteins largely remain unknown. Mutations in Rapsyn cause CMS, the most common mutation occurs at N88 since about 90% of Rapsyn-related CMS patients carry N88K mutation.

The authors demonstrate by genetic approaches and functional studies that the N88 residue of Rapsyn is essential for NMJ formation in mouse and provide novel mechanistic insights for Rapsyn function in postsynaptic differentiation. Rapsyn N88K mutation strongly impairs functional differentiation of NMJ as early as embryonic day 14. Evidence is presented that the mutation alters the E3 ligase activity of Rapsyn and MuSK-induced Rapsyn tyrosine phosphorylation. The authors further identify Y86 site as one of the main residues phosphorylated by MuSK and show that it is critical for Rapsyn E3 ligase activity and AChR clustering presumably by promoting Rapsyn self-association. Finally, rescue experiments using AAV-mediated Rapsyn expression in N88K mutant muscle indicate that WT-Rapsyn, but not Y86F-Rapsyn ameliorate the NMJ morphological deficits of N88K mutant thigh muscle suggesting a critical role for Y86 in Rapsyn function during NMJ formation. This study provides important mechanistic insights of the molecular mechanisms underlying Rapsyn function in NMJ formation. Although overall the data support the conclusions, some of the results and mechanisms proposed appear over-interpreted and care should be given to the phrasing all throughout the manuscript for accuracy.

1) The authors claim that their study led to discover a pathway by which signal is transduced from MuSK to Rapsyn. However, the sequential link between MuSK activation, Rapsyn self-association, E3 ligase activity and Rapsyn tyrosine phosphorylation events in AChR clustering is not clear and no evidence is provided to support the sequential model proposed by the authors in the discussion that may only reflect one possibility among many others. The author should clarify their signaling model and propose experimental data that support their conclusions as to the mechanistic frame of Rapsyn function in AChR clustering.

2) The authors demonstrate that Rapsyn N88K mutation impairs NMJ formation in vivo as early as E14, where only minimal agrin/MuSK signaling is occurring. It would be important to determine whether Rapsyn is tyrosine phosphorylated by MuSK at this early stage and discuss how MuSK leads to Rapsyn tyrosine phosphorylation and self-association in absence of agrin.

3) N88K mutation is located in the third TPR repeat known to promote Rapsyn oligomerization. However, data previously reported that the N88K mutation does not alter the ability of Rapsyn to self-associate in heterologous cells (Ohno et al., 2002) in contrast to the authors findings. This should be included and commented in the manuscript.

4) CMS patients with Rapsyn N88K mutation are either homozygous for N88K or heteroallelic for N88K and a second mutation in the Rapsyn coding region. Strikingly, clinical reports suggest that disease severity of patients with Rapsyn compound allelic mutations is greater than that of patients with homozygous N88K Rapsyn mutation. The authors should discuss their findings based on available clinical data on the effect of N88K mutation on the variability and severity of Rapsyn-related CMS.

5) A well-known function of Rapsyn is to anchor postsynaptic proteins in the subsynaptic cytoskeleton including AChR. Interactions between Rapsyn with microtubule and actin binding proteins have been reported in the literature. It is somehow surprising that the authors did not investigate (immunostaining) whether N88K mutation impairs Rapsyn association with cytoskeleton components.

Figure 1: The authors analyzed the NMJ phenotype of mice carrying Rapsyn N88K mutation. For a meaningful comparison of the NMJ phenotypes it is important to compare WT, heterozygous and homozygous mice for the mutation. To support the conclusions of the NMJ staining and EM data, the authors could perform ex vivo electrophysiological recording at P0 to evaluate the neurotransmission defects. This may be challenging since the mutant mice died at birth, but worth to try.

Figure 2: The staining of AChR clusters at E14 is of poor quality. The authors should provide a more suitable confocal image that reflects their quantification (almost no AChR clusters are visible in the mutant).

Fig5C: The authors should explain why a 2H treatment of agrin was used to evaluate the effect of N88K mutation in Rapsyn tyrosine phosphorylation in muscle cells.

*Reviewer #3:*

This paper describes the effect of the most frequent mutation (N88K) in the gene coding for the adaptor molecule rapsyn when tested in mice. The authors find that mice expressing the N88K-mutated rapsyn die within 24 hours after birth because of severe perturbation of the neuromuscular junction. The authors then show that N88K-rapsyn affects neddylation, rapsyn self-association and rapsyn-phosphorylation. Interestingly, prevention of phosphorylation of the nearby tyrosine (Y86) has a very similar effect as the N88K mutant.

Overall, the paper is well done and it reports on an interesting mechanism. Having said this, I find it a bit disturbing that others have reported much milder effects of the N88K mutant (see for example Cossins et al., 2006) in similar cell assays. These contributors also reported that the N88K-mutant rapsyn affects the stability of agrin-induced AChR clusters. Finally, the N88K mutant also has a much milder phenotype in human patients. Hence, a more thorough discussion and assessment what could account for these differences would strengthen the paper. One possibility that could explain the difference in the severity might be that levels of the N88K rapsyn mutant were much lower in the current study than in those studies conducted by others. Hence, a proper dose-response comparison between mutant and wild-type rapsyn may help address this difference.

---

## [Author Response]

Essential revisions:Reviewer #1:The neuromuscular junction has for decades been a model synapse for investigating basic mechanisms of synaptic transmission and synapse development.[…] The data presented are well documented, except that numbers of items examined (AChR clusters, animals) should be given (see e.g. (4.) below). Validations by appropriate controls are convincing,.

We thank Dr. Brenner for his constructive comments that have significantly improved the manuscript.

Reviewer #2:Essential revisions:[…] The authors demonstrate by genetic approaches and functional studies that the N88 residue of Rapsyn is essential for NMJ formation in mouse and provide novel mechanistic insights for Rapsyn function in postsynaptic differentiation. Rapsyn N88K mutation strongly impairs functional differentiation of NMJ as early as embryonic day 14. Evidence is presented that the mutation alters the E3 ligase activity of Rapsyn and MuSK-induced Rapsyn tyrosine phosphorylation. The authors further identify Y86 site as one of the main residues phosphorylated by MuSK and show that it is critical for Rapsyn E3 ligase activity and AChR clustering presumably by promoting Rapsyn self-association. Finally, rescue experiments using AAV-mediated Rapsyn expression in N88K mutant muscle indicate that WT-Rapsyn, but not Y86F-Rapsyn ameliorate the NMJ morphological deficits of N88K mutant thigh muscle suggesting a critical role for Y86 in Rapsyn function during NMJ formation. This study provides important mechanistic insights of the molecular mechanisms underlying Rapsyn function in NMJ formation. Although overall the data support the conclusions, some of the results and mechanisms proposed appear over-interpreted and care should be given to the phrasing all throughout the manuscript for accuracy.

We thank the reviewer for their comments “This study provides important mechanistic insights on the molecular mechanisms underlying Rapsyn function in NMJ formation” and constructive critiques and suggestions that have significantly improved the manuscript.

1) The authors claim that their study led to discover a pathway by which signal is transduced from MuSK to Rapsyn. However, the sequential link between MuSK activation, Rapsyn self-association, E3 ligase activity and Rapsyn tyrosine phosphorylation events in AChR clustering is not clear and no evidence is provided to support the sequential model proposed by the authors in the discussion that may only reflect one possibility among many others. The author should clarify their signaling model and propose experimental data that support their conclusions as to the mechanistic frame of Rapsyn function in AChR clustering.

This is an insightful question. We have performed additional experiments to characterize the time courses of MuSK activation, Rapsyn tyrosine phosphorylation, Rapsyn self-association, Rapsyn E3 ligase activity in C2C12 myotubes in response to Agrin stimulation. As shown in revised Figure 6A, 6B and Figure 7I, 7J, MuSK activation occurred within 10 minutes, followed by Rapsyn tyrosine phosphorylation and self-association (CO-IP of Rapsyn-HA and Rapsyn-EGFP in C2C12 cells) (which plateaued around 20 and 90 minutes, respectively). E3 ligase activity was not peaked until 90 min after Agrin stimulation (Figure 6B). These data demonstrate that these events occur sequentially. Together with other data presented in this paper, they support a working model that MuSK stimulates AChR cluster formation by increasing Rapsyn phosphorylation and self-association and thus enhancing its E3 ligase activity. These results are now presented in subsection “Reduced Rapsn tyrosine phosphorylation by N88K mutation” and subsection “Y86 phosphorylation for E3 ligase activity and AChR clustering”.

2) The authors demonstrate that Rapsyn N88K mutation impairs NMJ formation in vivo as early as E14, where only minimal agrin/MuSK signaling is occurring. It would be important to determine whether Rapsyn is tyrosine phosphorylated by MuSK at this early stage and discuss how MuSK leads to Rapsyn tyrosine phosphorylation and self-association in absence of agrin.

As suggested, we looked at Rapsyn tyrosine phosphorylation in E14 muscles. As shown in Figure 6E, 6G, tyrosine-phosphorylation was detected in muscles of E14 WT mice; but the level was reduced in muscles from N88K mice. As demonstrated in the literature, both MuSK kinase activity and Rapsyn are necessary for prepatterning or the formation of aneural AChR clusters (Lin et al., 2001; Yang et al., 2001b); and Rapsyn is known to interact with MuSK in the absence of Agrin (Apel et al., 1997). Further, LRP4 could interact with MuSK and thus maintain its activity in the absence of Agrin (Kim et al., 2008; Zhang et al., 2008). Our data are in agreement with these observations. Together, they suggest that tyrosine phosphorylation of Rapsyn could occur in the absence of Agrin and contributes to the formation of aneural AChR clusters or muscle prepatterning. These points are now described in the revised subsection “Reduced Rapsn tyrosine phosphorylation by N88K mutation” and Discussion section.

3) N88K mutation is located in the third TPR repeat known to promote Rapsyn oligomerization. However, data previously reported that the N88K mutation does not alter the ability of Rapsyn to self-associate in heterologous cells (Ohno et al., 2002) in contrast to the authors findings. This should be included and commented in the manuscript.

Good point. However, we would like to point out that in the paper by Ohno and colleagues (2002), “self-association” was defined as a morphological event (i.e., aggregate formation). In fact, as shown in Figure 4A, N88K Rapsyn could form aggregates in cultured cells, in agreement with the previous work. However, the N88K mutation reduced the co-precipitation of Rapsyn-HA and Rapsyn-GFP, suggesting that N88 is critical to self-association (Figure 7K-N). We revised subsection “Impaired ability of N88K Rapsn in AChR clustering” to indicate that our results were in agreement with the previous report in terms of “aggregates” formation.

4) CMS patients with Rapsyn N88K mutation are either homozygous for N88K or heteroallelic for N88K and a second mutation in the Rapsyn coding region. Strikingly, clinical reports suggest that disease severity of patients with Rapsyn compound allelic mutations is greater than that of patients with homozygous N88K Rapsyn mutation. The authors should discuss their findings based on available clinical data on the effect of N88K mutation on the variability and severity of Rapsyn-related CMS.

Good point. This point is now discussed in revised Discussion section.

CMS patients carrying the N88K mutation show variable phenotypes, from serious, early-onset symptoms to mild, late-onset symptoms (Cossins et al., 2006; Milone et al., 2009; Muller et al., 2003), even in patients with homozygous N88K mutation. In severe cases, N88K heteroallelic with c.966 + 1GT, L14P, or a frameshift mutation may cause postnatal death (Maselli et al., 2003; Milone et al., 2009; Richard et al., 2003), indicating N88K mutation could have seriously detrimental effects on NMJ formation or maintenance in human subjects. Overall, the symptoms of CMS patients with homozygous N88K mutation are milder than that of patients carrying heteroallelic N88K mutation with a second mutation. These may be due to that second mutations may exert more harmful effects than that of N88K mutation. Actually, among these second mutations, coding sequence deletions or insertions are often observed, such as c.1177-1178del AA, p.V50-S55del, p.K373del, c.553-554insGTTCT (Milone et al., 2009), which may cause frameshift of coding sequence and totally disrupt Rapsn’s function.

We also note that phenotypes of N88K mt mice seem to be more severe than symptoms of CMS patients. Almost all N88K mt mice died within 24 hours of birth. A simple explanation of these observations may be genetic difference. Gene mutations identified in patients with various diseases such as Lesch-Nyhan syndrome, Lowe syndrome, galactosemia diseases often failed to reproduce clinic symptoms in mouse models (Elsea and Lucas, 2002). On the other hand, a knock-in mutation of AQP2, a gene implicated in hereditary non-X-linked nephrogenic diabetes insipidus, causes more severe phenotypes in mice than in human subjects (Yang et al., 2001a). Even in same species, effects of a mutation could have variable impacts on different strains of mice, for example, the Agrin mutation, Agrin^nm380^ (Bogdanik and Burgess, 2011). Furthermore, it is possible that some patients with the N88K mutation are unable to survive and thus be examined by physicians.

5) A well-known function of Rapsyn is to anchor postsynaptic proteins in the subsynaptic cytoskeleton including AChR. Interactions between Rapsyn with microtubule and actin binding proteins have been reported in the literature. It is somehow surprising that the authors did not investigate (immunostaining) whether N88K mutation impairs Rapsyn association with cytoskeleton components.

Good question. To this end, we used two methods – Rapsyn-Actin association

(Walker et al., 1984) and Triton extraction (Moransard et al., 2003). First, as shown in Figure 5B, the amount of Actin and Rapsyn associated with surface AChR was similar between N88K and WT myotubes, suggesting the mutation has little effect on actin association. Second, actin-anchored Rapsyn-AChR complexes are resistant to low concentration of detergents (Moransard et al., 2003). We found that the amount of Rapsyn that could be solubilized by Triton X-100 was similar between N88K and WT (Figure 5—figure supplement 1C and D). These results suggest that the N88K mutation may not impair Rapsyn association with Actin cytoskeleton components. These points are now described in subsection “Reduced E3 ligase activity of N88K mt Rapsn”.

Figure 1: The authors analyzed the NMJ phenotype of mice carrying Rapsyn N88K mutation. For a meaningful comparison of the NMJ phenotypes it is important to compare WT, heterozygous and homozygous mice for the mutation. To support the conclusions of the NMJ staining and EM data, the authors could perform ex vivo electrophysiological recording at P0 to evaluate the neurotransmission defects. This may be challenging since the mutant mice died at birth, but worth to try.

As suggested, we have performed electrophysiological recordings to analyze synaptic transmission in four genotypes (WT, heterozygous N88K/+, homozygous N88K and N88K/-). As shown in Figure 3 and in subsection “Aberrant NMJ formation in N88K mt mice”, there was no difference in resting membrane potentials among the genotypes; however, fibers with mEPP, the amplitude and frequency of mEPPs were reduced in N88K mt and N88K/- mt, compared with WT or N88K/+. These results support the conclusions of morphological studies.

Figure 2: The staining of AChR clusters at E14 is of poor quality. The authors should provide a more suitable confocal image that reflects their quantification (almost no AChR clusters are visible in the mutant).

As suggested, better representative images are provided in revised Figure 2I-L.

Figure 5C: The authors should explain why a 2H treatment of agrin was used to evaluate the effect of N88K mutation in Rapsyn tyrosine phosphorylation in muscle cells.

Agrin activates MuSK within 10 minutes of stimulation and tyrosine phosphorylation of rapsyn that plateaus around 20 minutes (Figure 6A). Agrin induces the formation of AChR clusters beginning around 2 hours of stimulation (Ngo et al., 2012). Therefore, we studied the effect of N88K on Rapsyn tyrosine phosphorylation at 2 hours of agrin stimulation. We revised Materials and methods section to explain 2 hours treatment of Agrin for evaluating tyrosine phosphorylation of Rapsyn (subsection “C2C12 culture and Generation of Gene-modified C2C12 cells”).

Reviewer #3:[…] Overall, the paper is well done and it reports on an interesting mechanism. Having said this, I find it a bit disturbing that others have reported much milder effects of the N88K mutant (see for example Cossins et al., 2006) in similar cell assays. These contributors also reported that the N88K-mutant rapsyn affects the stability of agrin-induced AChR clusters. Finally, the N88K mutant also has a much milder phenotype in human patients. Hence, a more thorough discussion and assessment what could account for these differences would strengthen the paper. One possibility that could explain the difference in the severity might be that levels of the N88K rapsyn mutant were much lower in the current study than in those studies conducted by others. Hence, a proper dose-response comparison between mutant and wild-type rapsyn may help address this difference.

We thank Dr. Ruegg for his constructive comments that have significantly improved the manuscript.

Regarding “much milder effects of the N88K mutant (see for example Cossins et al., 2006) in similar cell assays” – These points are now discussed in the revised Discussion section –. Pathological mechanisms of the N88K mutation have been examined previously. It was shown to reduce Rapsn’s ability to induce AChR clusters in HEK and TE671 cells (Cossins et al., 2006; Ohno et al., 2002). In agreement, the N88K mutant was shown here to reduce Rapsn’s ability to induce AChR clusters in HEK293T cells (Figure 4A). In N88K knock-in mt mice and in CRISPR/Cas9-generated C2C12 myotubes, AChR clusters were almost diminished.

However, N88K seemed to be able to induce AChR clusters in *Rapsn* mt myotubes (Cossins et al., 2006). The cause of this discrepancy is unclear. In the current study, the impact of the N88K mutation was inferred mainly by loss-of-function approaches, both in vitro and in vivo, without changing endogenous Rapsn level (Figure 1 and Figure 4E and F). Rapsn is known to induce AChR clusters at low concentrations, but inhibit AChR clusters at high level (Han et al., 1999; Yoshihara and Hall, 1993). This inverted U-function could complicate the interpretation of results of overexpressing WT or mt Rapsn. These points are now discussed in the revised Discussion section.

Regarding “much milder phenotypes in human patients” –These points are now discussed in the revised Discussion section –. CMS patients carrying the N88K mutation show variable phenotypes, from serious, early-onset symptoms to mild, late-onset symptoms (Cossins et al., 2006; Milone et al., 2009; Muller et al., 2003), even in patients with homozygous N88K mutation. In severe cases, N88K heteroallelic with c.966 + 1GT, L14P, or a frameshift mutation may cause postnatal death (Maselli et al., 2003; Milone et al., 2009; Richard et al., 2003), indicating N88K mutation could have seriously detrimental effects on NMJ formation or maintenance in human subjects. Overall, the symptoms of CMS patients with homozygous N88K mutation are milder than that of patients carrying heteroallelic N88K mutation with a second mutation. These may be due to that second mutations may exert more harmful effects than that of N88K mutation. Actually, among these second mutations, coding sequence deletion or insertion are often observed, such as

c.1177-1178del AA, p.V50-S55del, p.K373del, c.553-554insGTTCT (Milone et al., 2009), which may cause frameshift of coding sequence and totally disrupt Rapsn’s function. We also note that phenotypes of N88K mt mice seem to be more severe than symptoms of CMS patients. Almost all N88K mt mice died within 24 hours of birth. A simple explanation of these observations may be genetic difference. Gene mutations identified in patients with various diseases such as Lesch-Nyhan syndrome, Lowe syndrome, galactosemia diseases often failed to reproduce clinic symptoms in mouse models (Elsea and Lucas, 2002). On the other hand, a knock-in mutation of AQP2, a gene implicated in hereditary non-X-linked nephrogenic diabetes insipidus, causes more severe phenotypes in mice than in human subjects (Yang et al., 2001a). Even in same species, effects of a mutation could have variable impacts on different strains of mice, for example, the Agrin mutation, Agrin^nm380^ (Bogdanik and Burgess, 2011). Furthermore, it is possible that some patients with the N88K mutation are unable to survive and thus be examined by physicians.